# Novel Eye Drop Delivery Systems: Advance on Formulation Design Strategies Targeting Anterior and Posterior Segments of the Eye

**DOI:** 10.3390/pharmaceutics14061150

**Published:** 2022-05-27

**Authors:** Yaru Wang, Changhong Wang

**Affiliations:** Institute of Chinese Materia Medica, Shanghai University of Traditional Chinese Medicine, The MOE Key Laboratory for Standardization of Chinese Medicines, Shanghai R&D Centre for Standardization of Chinese Medicines, 1200 Cailun Road, Shanghai 201203, China; 15256089844@163.com

**Keywords:** topical administration, traditional eye drops, novel eye drop delivery systems, anterior segment, posterior segment

## Abstract

Eye drops are the most common and convenient route of topical administration and the first choice of treatment for many ocular diseases. However, the ocular bioavailability of traditional eye drops (i.e., solutions, suspensions, and ointments) is very low because of ophthalmic physiology and barriers, which greatly limits their therapeutic effect. Over the past few decades, many novel eye drop delivery systems, such as prodrugs, cyclodextrins, in situ gels, and nanoparticles, have been developed to improve ophthalmic bioavailability. These novel eye drop delivery systems have good biocompatibility, adhesion, and propermeation properties and have shown superior performance and efficacy over traditional eye drops. Therefore, the purpose of this review was to systematically present the research progress on novel eye drop delivery systems and provide a reference for the development of dosage form, clinical application, and commercial transformation of eye drops.

## 1. Introduction

The eye is a delicate and complicated organ with a unique anatomical and physiological structure that can be divided into two main parts: the anterior and posterior segments. The anterior segment includes the cornea, iris, ciliary body, pupil, and lens, while the posterior segment consists of the vitreous humor, sclera, choroid, and retina (Figure 1) [1]. Common diseases occurring in the anterior segment include keratitis, dry eye, cataract, glaucoma, inflammatory diseases, and infection diseases [2]. Diseases affecting the posterior segment include age-related macular degeneration (AMD) [3], diabetic retinopathy macular edema (DME), proliferative vitreoretinopathy (PVR), and cytomegalovirus (CMV) [4,5]. In clinical practice, systemic administration, vitreous injection, periocular injection, and topical administration routes (Figure 1) are commonly used to treat ocular diseases [6]. Systemic administration is often used to treat the anterior and posterior segments diseases of the eye. However, the ocular bioavailability of this route is extremely low (<2%) because of the presence of blood–eye barriers [7]. In addition, this route is frequently associated with systemic toxicity and serious side effects due to large doses. Vitreous injection and periocular injection are commonly applied to treat diseases of the posterior segment of the eye. In these methods, the drug is delivered directly around the lesion, allowing higher concentrations to be maintained and avoiding systemic exposure [8]. However, as an invasive procedure, vitreous injection may cause discomfort and eye pain. In addition, the eye is easily affected by side effects, such as vitreous detachment, retinal hemorrhagic inflammation, and increased intraocular pressure in the case of repeated injection [9]. Although periocular injection is a less invasive method than vitreous injection, the drug still needs to overcome some biological barriers (conjunctiva, sclera, and choroid) to reach the site of action [10]. Topical administration is commonly used to treat diseases of the anterior segment. Compared with other routes of administration, topical administration is the least invasive method and has the highest patient compliance [11]. Moreover, topically administered drugs have the advantages of being easy to manufacture and having low cost and low side effects, which make topical administration the most desirable clinical application route. Drugs administered through the topical route are usually formulated into eye drops, which account for 90% of the commercialized products in the global ophthalmic drug market, with ophthalmic solutions, suspensions, and ointments accounting for 62.4%, 8.7%, and 17.4%, respectively [12]. Ophthalmic solutions tend to drain rapidly from the conjunctival sac, which allows little time for the drug to enter the ocular tissues. The dissolution characteristics of the drug in the suspension are poor, as the volume size of the drops is generally in the range of 25–50 μL, which prevents the drug from forming a high concentration gradient in the ocular tissues [13]. Ointments are not commonly used to treat ocular diseases, because the oil components can affect vision for some time after application. The absorption of these traditional eye drops on the outer surface of the eye is limited by different barriers such as static (corneal and blood–aqueous barrier (BAB)), dynamic (tear drainage, conjunctival blood, and lymphatic flow), and metabolic barriers [11], making the bioavailability of the drug after topical administration very low (<5% of the administered dose) [14] and difficult to be delivered to the posterior segment of the eye.

In recent years, a number of advanced drug delivery systems have been developed to overcome the limitations of traditional eye drops. These novel drug delivery systems have shown promising results in in vivo and in vitro ocular disease models as well as in clinical practice by increasing the retention time of drugs in front of the cornea, promoting drug penetration, or facilitating drug delivery to the eye through the corneal/conjunctival–scleral pathway. Research on novel drug delivery systems has now been extensively reported. For example, Jumelle et al. reviewed the progress and limitations of in situ gels and nanoparticle drops [15]. Yellepeddi and Palakurthi combed through the progress of in situ gels and nanoparticle drops [16]. Alvarez-Trabado et al. summarized the progress of lipid nanoparticles for topical drug delivery design [17]. Clolkar et al. summarized new strategies for ocular drug delivery in the anterior segment [18]. Madni et al. reviewed noninvasive strategies for the posterior segment of the eye [19]. In contrast to these studies, we do not only present a comprehensive overview of the results achieved in the anterior and posterior segments of the eye following the administration of prodrug, cyclodextrin, in situ gel, and nanoparticle delivery systems in the form of eye drops. More importantly, we also investigate the high-ocular-bioavailability eye drops based on these novel delivery systems, such as prodrug-based self-assembled nanoparticle eye drops, cyclodextrin-based prodrug eye drops, highly adherent eye drops, nanoparticle-based cationic nanoparticle eye drops, surface-modified nanoparticle eye drops, adherent nanoparticle eye drops, and ligand-targeted nanoparticle eye drops. The study of novel eye drop delivery systems in the anterior and posterior segments of the eye may provide some basis for the selection of dosage forms, clinical application, and commercial translation of eye drops.

## 2. Physiological and Structural Barriers of the Eye

When the drug passes through the tear film and reaches its target site, it may encounter a variety of dynamic or static barriers present in tissues, such as the cornea barrier, conjunctiva barrier, blood–aqueous barrier, and blood–retina barrier (BRB) (Figure 1).

### 2.1. Tear Film Barrier

The tear film is the first obstacle faced for topically administered drugs. The precorneal volume is limited, and the maximum volume of eye drops that can be held in the conjunctival sac is approximately 30 µL. After topical application of eye drops (25–50 µL), only about 10 µL of the drug could remain because of the blink response and drainage from the nasolacrimal duct [20]. The drug that remains in front of the cornea mixes with the tear film secreted by the lacrimal glands and cupped cells. An important obstacle facing the drug at this point is tear turnover. Normally, the tear film has a volume of approximately 7–9 μL, a thickness of approximately 7–10 μm, and a turnover rate between 0.5 and 2.2 μL/min [14]. The sudden increase in tear film volume leads to an increase in tear film turnover rate and rapid clearance of drug molecules through tear drainage within minutes. In addition, the tear film consists of an external lipid layer, an intermediate aqueous layer, and an internal mucin layer [21]. The lipid and aqueous layers restrict the absorption of hydrophilic and hydrophobic drugs, respectively. The negatively charged mucin layer in the inner tear layer prevents negatively charged drugs or carriers from entering the cornea.

### 2.2. Corneal Barrier

The cornea is the second ocular barrier limiting the penetration of exogenous substances into the eye. It consists of five collagenous layers: the epithelium, Bowman’s membrane, the stroma, Descemet’s membrane, and the endothelium [22]. Bowman’s membrane and Descemet’s membranes are noncellular structural elastic layers composed of collagen and glycoprotein. The layers that form substantial barriers to drug penetration are the epithelium, stroma, and endothelium. The superficial corneal epithelium makes up six to eight layers of cells with a total thickness of approximately 40–50 µm. The epithelium cell becomes flattened during maturation and eventually forms tight intercellular junctions, which allow the permeation of hydrophilic drugs at a rate of only 10^−7^–10^−5^ cms^−1^ [14]. The hydrophilic matrix next to the corneal epithelium (approximately 80% water content) has a thickness of approximately 450–500 µm, representing 90% of the corneal thickness, and in turn imposes significant limitations on lipophilic drugs due to solubility and partition coefficients [23]. Likewise, tight junctions exist on the endothelium. However, the endothelium is leakier than the epithelium and offers less resistance to paracellular drug transport because of its small cell thickness (13 µm). Overall, the specific sandwich structure of corneal tissue makes it a unique barrier for most lipophilic and hydrophilic drugs. In addition, efflux pumps such as P-glycoprotein (P-gp), multidrug resistance-associated protein (MRP), and breast cancer resistance protein (BCRP) [24] expressed on epithelial and endothelial cells are important barriers to drug absorption.

### 2.3. Conjunctival Barrier

The conjunctiva is a thin, translucent, vascularized mucous membrane that can be divided into three parts: the bulbar conjunctiva, the conjunctival vault, and the lid conjunctiva. It is made up of three layers: the outer epithelium, the lamina propria, and the submucosa. The conjunctiva, unlike the cornea, is considered an important route for noncorneal drug delivery (e.g., macromolecular nanoparticles) because of its rich vascular system, cupped cells, and potential for transdifferentiation. The surface area of the conjunctiva is 17 times greater than that of the cornea [25], which allows higher uptake to occur in this tissue. The conjunctival epithelial cell gap is wider than the corneal epithelial cell gap, so hydrophilic macromolecules are generally more permeable in the conjunctiva than in the cornea. However, because of the presence of conjunctival capillaries and lymph, drug absorption through the conjunctiva is prone to a significant loss in the body’s circulation, thereby reducing overall ocular bioavailability [26]. In addition, the conjunctival epithelium expresses a variety of transporters, such as neutral and cationic amino acid transporters (ATB^0,+^), nucleoside transporters (CNT_2_), peptide transporters (PepT_1_), P-gp and MRP [27,28,29], which also have an impact on drug absorption.

### 2.4. Blood–Aqueous Barrier

The BAB is a barrier located anteriorly in the eye, formed by the vascular endothelium of the iris and ciliary muscle and the nonpigmented epithelium at the back of the iris [30]. The barrier is hypopermeable because of the tight junctions between the two cell layers. The BAB not only limits the passage of drugs from the blood to or from the atrial aqueous but further restricts the concentration and retention time of drugs in the atrial aqueous because of the constant drainage of the atrial aqueous (turnover rate of 2.0–3.0 mL/min) [31].

### 2.5. Blood–Retinal Barrier

The BRB is a barrier located in the posterior segment of the eye that consists of two types of cells: retinal capillary endothelial cells (RPCs, internal BRB) and retinal pigment epithelial cells (RPEs, external BRB). Both RPCs and PREs are considered to be important barriers to retinal drug transport because of the presence of tight junctions therein [32]. In addition, the pump proteins and internal trapped vesicles (e.g., Na^+^/K^+^-ATP, P-gp, MRP_1_) [33] expressed in RPE cells allow only specific active drugs to be exchanged between the choroid and the retina [34].

## 3. Drug Absorption Routes of Eye Drops

After administration of eye drops, the drug reaches the intraocular tissues mainly via the corneal/conjunctival–scleral route (Figure 2) [35]. These absorption routes are governed by the physicochemical properties of the drug, the form of administration, and the physiological structure and barriers of the eye [36,37]. The corneal route is the primary route of topical drug delivery. When the drug is dropped onto the surface of the eye, it reaches the corneal epithelium via the cellular bypass (hydrophilic drug) and transcellular (lipophilic drug) routes and then reaches the anterior chamber. Once the drug crosses the cornea to reach the anterior chamber, it is distributed to the surrounding tissues (lens, iris, ciliary body). At this point, there are two main pathways for the drug to reach the posterior segment of the eye: (1) direct diffusion to the tissues of the vitreous, retina, choroid, and sclera via the transvitreal route and (2) drainage to the posterior segment of the eye via the uveal–scleral pathway [38].

The conjunctival–scleral route means that after absorption through the conjunctiva, the drug can cross the sclera into the choroid, retina, or optic nerve tissues [39]. Because of the presence of a rich capillary network and lymph nodes in the conjunctiva, transconjunctival absorption readily allows the drug to enter the systemic circulation, thereby reducing the concentration in the posterior tissues of the eye. Nevertheless, because of the large absorption surface area and the relatively wide cellular gap in the conjunctiva, some compounds with large molecular weights or poor corneal permeability can still reach the intraocular tissues through the conjunctiva and sclera by passive diffusion or active transport routes.

In addition, a very small amount of drugs can reach the tissues of the anterior or posterior segments of the eye by systemic absorption via the nasolacrimal duct, conjunctival vessels, and lymphatic vessels [40].

## 4. Novel Eye Drop Delivery Systems

Because of the unique physiological structure and barrier of the eye, the ocular bioavailability of traditional eye drops is very poor. Novel eye drop delivery systems such as prodrugs, cyclodextrins, in situ gels, and nanoparticles have been developed to improve the bioavailability of drugs in the eye. These novel eye drop delivery systems have shown better clinical results than conventional eye drops in the treatment of both anterior and posterior segments of the eye.

### 4.1. Prodrug Eye Drop Delivery Systems

Prodrugs are inactive compounds obtained by chemical modification of the active compounds through esters, carbamates, phosphates, and oximes, which can be converted to their active compounds in vivo by chemical or enzymatic metabolic processes [41]. Prodrugs are endowed with new physicochemical and biochemical properties from the active drug. The most important advantages of using prodrug designs are enhancing transmembrane permeation of the drug in ocular tissues and improving ocular bioavailability with minimal disruption of the ocular barrier [42].

The concept of prodrugs was introduced into the field of ophthalmology in 1976 by Hussain et al. [43] to enhance the uptake of the highly polar molecule epinephrine through lipid membranes. Since then, various prodrugs have been designed to improve the physicochemical properties of therapeutic drugs. Because of the high expression of esterases in the corneal epithelium, the corneal drug concentration of the optimal ester prodrug, dipivalyl epinephrine (bipivalin), was 17 times higher than that of epinephrine [44]. Brinzolamide, a commercial glaucoma treatment drug, when prepared as brinzolamide prodrug eye drops, penetrated corneal tissue more easily than commercial brinzolamide eye drops and were more effective in reducing intraocular pressure (IOP). More importantly, the prodrug eye drops were not toxic to the corneal epithelium after 1 week of repeated administration [45]. In addition, Babizhayev found that *N*-acetylcarnitine (NAC) prodrug eye drops could deliver more of the active ingredient L-carnitine to the aqueous humor and lens tissues [46].

Prodrugs also enable more of the drug to be delivered to the posterior segment of the eye. For example, not only did resolvyx (RX-10045) micelle have sufficient therapeutic concentrations in the anterior segment, but its active metabolite resolving E1 analogue (RX-10008) was visible in the retinal/choroidal tissue after topical administration [47]. When supplied in the form of eye drops, 4-chloro-3-(5-methyl-3-{[4-(2-pyrrolidin-1-ylethoxy)phenyl]amino}-1,2,4-benzotriazin-7-yl)phenyl benzoate, a prodrug with thermal stability, was readily converted to 4-chloro-3-{5-methyl-3-[4-(2-pyrrolidin-1-yl-ethoxy)phenylamino]benzo[1,2,4]triazin-7-yl}phenol, an active compound that could inhibit the VEGFr2 and Src family (Src and YES) kinase signaling pathways for the treatment of AMD [48].

Amphoteric prodrugs can spontaneously self-assemble into nanoparticles with increased stability, solubility, and ocular bioavailability [49]. Hu et al. prepared a self-assembled nanoparticle using the prodrug paclitaxel–peptide amphiphilic compound (PS-GA-RGD). The release results showed that active paclitaxel (PS) was continuously released from PS-GA-RGD nanoparticles within 48 h. In an in vivo eye stimulation test, the PS-GA-RGD nanoparticle was well tolerated in the eye after a single instillation. More importantly, the efficacy of the prodrug nanoparticles was better than that of PS in the treatment of dry eye [50]. In addition, Stella et al. obtained a lipid drug conjugate (LDC), 4′-trisnorsqualenoylacyclovir (SQACV), by covalently linking the 4′-hydroxyl group of acyclovir (ACV) to the isoprene chain of squalene, which was subsequently formulated into self-assembled nanoparticles using a nanoprecipitation method. The pharmacokinetic characterization in rabbit tear fluid and aqueous humor showed that the SQACV self-assembled nanoparticles increased the ACV content in rabbit aqueous humor compared with the free ACV eye drops [51]. Table 1 [45,46,52,53] lists prodrug eye drop delivery systems in the application of the anterior and posterior segments of the eye.

### 4.2. Cyclodextrin (CD) Eye Drop Delivery Systems

Cyclodextrins (CDs) are a class of natural cyclic oligosaccharides. The most common natural CDs consist of 6 (α-CD), 7 (β-CD), and 8 (γ-CD) 1,4-linked α-D-glucopyranose units (Figure 3). The lipophilic inner cavity of CDs is surrounded by a hydrophilic outer surface, which predetermines its ease of encapsulation of water-insoluble chemicals by noncovalent conjugation (electrostatic interactions, van der Waals contributions, hydrogen bonding, and charge transfer interactions). CDs allow enhancing drug permeability and biocompatibility and reducing irritation and are widely used in ophthalmic formulations. In addition, the special chemical structure of CDs can improve the stability, dispersion, and dissolution properties of certain drugs and significantly enhance their physical and chemical activity [54].

CDs have been increasingly studied in ocular drug delivery systems in recent decades. To improve the solubility and ocular bioavailability of the active imidazole antifungal drug econazole nitrate (EC), EC-β-CD and EC hydroxypropyl-β-cyclodextrin (EC-HP-β-CD) inclusion complexes were prepared using the coprecipitation and freeze-drying techniques, respectively. The two inclusion complexes increased the solubility of EC by approximately threefold and fourfold, respectively. Furthermore, in vivo experiments showed that eye drops containing EC-CD inclusion complexes were more bioavailable than EC eye drops [55].

CD eye drop delivery systems also make it possible to deliver the drug to the posterior part of the eye. Loftsson et al. found that the concentration of dexamethasone (DEX) in aqueous humor was 66 ± 20 ng/g after 2 h of topical DEX ophthalmic solution. For 1.3% *w*/*v* DEX/HM-β-CD, the value was determined to be 320 ± 230 ng/g [56]. After a single application of 1.5% *w*/*v* DEX/γ-CD for 2 h, the concentration of DEX was 236 ± 67 ng/g in the aqueous humor, 29 ± 16 ng/g in the vitreous humor, and 57 ± 22 ng/g in the retina [57]. After 2 h of administration of 0.5 and 1.5% *w*/*v* DEX/randomly methylated β-cyclodextrin (RM-β-CD) eye drops, the aqueous humor DEX levels were 1190 ± 110 and 1670 ± 630 ng/g, respectively. Levels in the retina were 33 ± 7 and 66 ± 49 ng/g, and those in the optic nerve were 41 ± 12 and 130 ± 50 ng/g, respectively [56]. The above data suggest that γ-CD, hydrophilic HM-β-CD, and lipophilic RM-β-CD can enhance the topical transport of DEX in the eye. Furthermore, the reason why lipophilic RM-β-CD resulted in higher DEX concentrations than HM-β-CD and γ-CD may be the fact that lipophilic RM-β-CD not only enhances drug delivery to the lipophilic cornea and sclera through the aqueous tear film but reduces the tear film’s barrier function by penetrating the membrane.

CDs can improve the solubility and stability of insoluble prodrugs and reduce their irritation, making them more suitable for ocular administration. For example, latanoprost is a practically insoluble prostaglandin F2α analog considered a first-line agent for glaucoma treatment. Its poor aqueous solubility makes it challenging to formulate into eye drops. Hu et al. found that propylaminoβCD could effectively form a complex with latanoprost, and the complex formed not only protected the ester bond of latanoprost but promoted the dissolution of latanoprost. Furthermore, in vivo experiments demonstrated that the latanoprost propylaminoβCD formulation led to lower ocular irritation than the commercial latanoprost formulation used as a reference. The latanoprost propylaminoβCD formulation was demonstrated to successfully address the main stability, solubility, and tolerance limitations of topical ocular latanoprost therapy for glaucoma [58].

The low viscosity of CDs makes them easy to remove from the anterior cornea, so the addition of cellulose derivatives to CDs solutions or in combination with in situ gels is an effective strategy to improve the ocular bioavailability of the drug [59,60]. Ref. [61] showed that fluconazole HP-β-CD inclusion complexes hydrogels not only had good bioadhesive properties but were effective in controlling the release of fluconazole. Ketoconazole (KTZ) complexes with sulfobutyl ether-β-cyclodextrin (SBE-β-CD) prepared by Chaudhari et al. showed a fivefold increase in solubility of KTZ compared with KTZ solution. After loading the KTZ-SBE-β-CD complex into in situ gels, KTZ exhibited more sustained release properties due to the diffusion of the gel, and the corneal retention and penetration of KTZ were significantly increased because of the adhesive properties of the gel [62]. Table 2 [55,56,57,63,64,65,66,67] lists cyclodextrin eye drop delivery systems in the application of the anterior and posterior segments of the eye.

### 4.3. In Situ Gel Eye Drop Delivery Systems

In situ gels consist of environmentally sensitive polymers that can rapidly transform a solution into a viscoelastic semisolid gel within the conjunctival sac in response to environmental stimuli such as pH, temperature, and ionic strength and finally slow release of a drug under physiological conditions (Figure 4) [68,69,70]. Environmental stimuli that affect in situ gels can be classified as biological, chemical, and physical stimuli. Biological stimuli include changes in glucose levels and enzyme activity; chemical stimuli involve changes in pH and ionic strength in biological fluids [71]; and physical stimuli include changes in temperature, sound, electric fields, pressure, light, and magnetic fields [72,73,74]. As a novel drug delivery system, in situ gels eye drops have many formulation advantages: (1) ease of use, accurate drug delivery, and good reproducibility; (2) prolonged retention of the drug at the desired site, reducing frequent administration; (3) simple production process and easy scalability to industrial frameworks; (4) the ability to deliver sensitive drugs [75]; and (5) good tolerability [76]. The most studied in situ gels are temperature-sensitive, pH-sensitive, and ion-sensitive in situ gels.

#### 4.3.1. Temperature-Sensitive In Situ Gel Eye Drop Delivery Systems

Temperature-sensitive in situ gels (abbreviated: temperature-sensitive gels) are among the longest studied, most widely used, and most common in situ gels. The temperature at which the solution–gel transition occurs is called the phase transition temperature, also known as the critical solution temperature (CST). The transition occurs at either the lower critical solution temperature (LCST) or the upper critical solution temperature (UCST), depending on the type of polymer used. There are three common types of temperature-sensitive gels, namely positive temperature-sensitive gels [77], negative temperature-sensitive gels, and thermally reversible gels [78]. Positive temperature-sensitive gels undergo a transition below UCST. Negative temperature-sensitive gels have LCST and shift when heated above LCST [79]. Thermally reversible gels can change from solution to gel as the temperature decreases and then convert from gel to solution when heated again. Regardless of the type of temperature-sensitive gel, the phase transition should occur at the precorneal temperature (35 °C) to avoid dilution of the lacrimal fluid and rapid elimination of the drop in front of the cornea after administration [80].

Polymers with temperature sensitivity are known as ON/OFF polymers [81] and mainly include poloxamer and poly(*N*-isopropyl acrylamide). A large number of studies on temperature-sensitive gel eye drop delivery systems have been reported in the last decades. A temperature-sensitive gel system loaded with brinzolamide formed a gel at 33.2 ± 1.1 °C with a diffusion-controlled release time of 8 h. In vivo studies showed that this temperature-sensitive gel had better drug retention than commercial formulations [82]. Likewise, a temperature-sensitive gel loaded with ketoconazole (KCL) formed a gel at 33 °C, and KCL was moderately released from the temperature-sensitive gel without a bursting effect. In vivo antimicrobial studies showed a higher healing effect of temperature-sensitive gels compared with that of commercial eye drops [83].

Cellulose derivatives, such as chitosan (CS) and hydroxypropyl methyl cellulose (HPMC), are known to improve the anterior corneal retention time and bioavailability of the in situ gels [84]. A study by Gratieri et al. showed that CS improved the mechanical strength and structural properties of poloxamer formulations and conferred mucoadhesive properties in a concentration-dependent manner. After 10 min of instillation of the poloxamer/CS 16:1 formulation in the human eye, 50–60% of the gel remained in contact with the corneal surface, while only 15% of the drug solution remained in contact with the cornea. This demonstrated a fourfold increase in retention of poloxamer/CS compared with that of conventional solutions. Therefore, the in situ forming gel comprising poloxamer/CS is a promising tool for the topical treatment of ocular diseases [85]. For temperature-sensitive in situ gels loaded with ketorolac aminotriol nanodispersions, decreasing the concentration of Pluronic F127 increased the gelation time and gelling temperature of in situ gels, and adding HPMC to Pluronic F12 hydrogels significantly improved the adhesion strength of the gels [86].

Salts are known to lower the temperature and regulate the release of in situ gels to maximize their efficacy following intraocular administration. A study investigating the effect of different salts on the properties of methylcellulose (MC)-based in situ gels found that 5–7% *w*/*v* sodium chloride, 8–9% *w*/*v* potassium chloride, and 5% *w*/*v* sodium bicarbonate were able to reduce the CST below the physiological temperature (37 °C) [87]. Rheological studies indicated a large increase in viscosity at 37 °C with the addition of salts in MC solutions, and the release time of the drug from salted MC solutions could be increased by 1.5 h to 3–5 h depending on the concentration and type of salt. Therefore, the salted MC solutions were a better alternative than the MC solution for enhancing the ocular bioavailability of the drug. Another study demonstrated that the use of carrageenan and potassium chloride was effective in lowering the CST of native MC solutions from 60 °C to 33.5 °C [88]. Table 3 [82,83,89,90] lists temperature-sensitive in situ gel eye drop delivery systems in the application of the anterior and posterior segments of the eye.

#### 4.3.2. pH-Sensitive In Situ Gel Eye Drop Delivery Systems

pH-sensitive in situ gels consist of pH-sensitive polymers, which are polyelectrolytes containing acidic groups (carboxylic or sulphonic acids) or basic groups (ammonium salts) that accept or release protons in response to changes in the pH of the surrounding environment. At a lower pH (pH 4.4), the formulation exists as a convenient solution, but it forms a gel at pH 7.4 (i.e., the pH of tears).

The main pH-sensitive polymers most commonly used in ophthalmic preparations are polyacrylic acid (PLA), methacrylic acid (MAc), *N*,*N*-dimethyl aminoethyl methacrylate (DMAEMA), cellulose derivatives, and cellulose phthalate acetate [91,92]. pH-sensitive in situ gel eye drop delivery systems have great potential for maintaining drug stability and release. For example, pH-sensitive in situ gel prepared using carbomer 974P as a gelling agent and HPMC as a viscosity builder was used for the sustained delivery of the ophthalmic drug baicalin. The results of rheological studies showed that the gel strength was significantly enhanced under physiological conditions and that the gel provided sustained release of the drug within 8 h. In addition, the area under the curve (AUC) and the plasma peak concentration (C_max_) values of the in situ gel were 6.1 and 3.6 times higher than those of the control solution, respectively [93]. Gupta et al. prepared a pH-sensitive in situ gel loaded with timolol maleate (TM) for the treatment of glaucoma using carbopol as a gelling agent and CS as a viscosity builder. The in situ gel was in a liquid state at room temperature and the formulated pH (pH 6.0) and rapidly transformed to a viscous gel phase at the pH of the tear (lacrimal) fluid (pH 7.4). The results of in vitro drug release and in vivo effects demonstrated that the in situ gel was therapeutically effective compared with Glucomol^®^ (0.25% TM ophthalmic solution), 0.4% *w*/*v* carbopol solution, and liposome formulations and exhibited a sickle-like (diffusion-controlled) release behavior within 24 h [94]. A pH-sensitive in situ gel loaded with gatifloxacin was prepared by Kanoujia et al. using carbopol 940 as a gelling agent and HPMC and HPMC K15M as viscosity builders. The optimum formulation is clear and transparent with pH, viscosity, and drug content in the ranges of 6.0–6.8, 10–570 cps, and 82–98%, respectively. In vitro release results showed that the gel provided sustained drug release over 8 h and that the mechanism of drug release from the gel was controlled by diffusion [95]. In addition, a ketorolac tromethamine-loaded pH-sensitive in situ gel showed pseudoplastic rheology, which was able to improve bioavailability through its longer corneal residence time and ability to maintain drug release. Also importantly, it was easy to perfuse and reduced the frequency of perfusion, thus making it more acceptable to the patient [96]. Table 3 [93,94,95,96,97,98,99] lists pH-sensitive in situ gel eye drop delivery systems in the application of the anterior and posterior segments of the eye.

#### 4.3.3. Ion-Sensitive In Situ Gel Eye Drop Delivery Systems

Ion-sensitive in situ gels form crosslinks with cations (Na^+^, Ca^2+^, Mg^2+^) present in the tear fluid to form a gel on the ocular surface and prolong corneal contact time. The most commonly used polymers in ion-sensitive in situ gel formulations include junctional cold gels, hyaluronic acid, and sodium alginate. Ion-sensitive in situ gel eye drop delivery systems are becoming more widely used in the treatment of ocular diseases [100]. A ketotifen-loaded ion-sensitive eye drop system prepared using natural polysaccharide, deacetylase conjugates had an optimal viscosity. After dropping into the eye, it underwent a rapid sol–gel transition due to ionic interactions. Furthermore, the viscosity of the formulation remained unchanged over a storage period of 180 days. More importantly, at the same dose, the in situ gel exhibited more durable pharmacological behavior than regular eye drops [101]. An ion-sensitive in situ gel loaded with terbinafine hydrochloride nanocolloids was transparent, pseudoplastic, and mucoadhesive. It not only released the drug slowly but improved the mean residence time and ocular bioavailability of the terbinafine [102]. Rupenthal et al. formulated ion-sensitive in situ gels based on junctional gum, xanthan gum, and carrageenan and evaluated these preparations for in vivo release, precorneal retention time, and ocular irritation. The results showed that the in situ gels were nonirritating, with 2.5-fold increases in the AUC and pupil constriction response of pilocarpine compared with aqueous solutions [103]. In addition, a carrageenan-based ion-sensitive in situ gel eye drop system was prepared using ACV as the model drug, HPMC as the mucoadhesive agent, and HP-β-CD as the permeation enhancer. At 2% HP-β-CD, the apparent permeation coefficient of ACV was approximately 2.16 times higher than that of conventional eye drops. The ion-sensitive in situ gel significantly delayed drug release and improved bioavailability compared with conventional eye drops [104]. Table 3 [101,102,105,106] lists ion-sensitive in situ gel eye drop delivery systems in the application of the anterior and posterior segments of the eye.

**Table 3 pharmaceutics-14-01150-t003:** In situ gel eye drop delivery systems in the application of the anterior and posterior segments of the eye.

Model Drugs	Indications	Main Findings	Ref.
**Anterior segment**			
**Temperature-sensitive in situ gels**			
Brinzolamide	Glaucoma	The optimal formulation formed a gel at 33.2 ± 1.1 °C with a diffusion-controlled release time of 8 h.	[82]
Ketoconazole	Eye infections	The temperature of the gel was 33 °C, and the gel had a higher healing effect than commercial eye drops.	[83]
Tetrahydrozoline	Conjunctivitis	The best prescriptions were stable, nonirritating, and provided continuous drug release for up to 24 h.	[89]
Dorzolamide	Glaucoma	The retention time of the drug in front of the cornea was prolonged, and bioavailability was improved.	[90]
**p** **H-sensitive in situ gels**			
Baicalin	Eye infections	In situ gel provided sustained release of the drug within 8 h.	[93]
Timolol maleate	Glaucoma	In situ gel eye drops rapidly transformed into a mucoadhesive gel at the pH of tears.	[94]
Gatifloxacin	Eye infections	Gel provided drug release over 8 h.	[95]
Ketorolac tromethamine	Eye infections	The retention time of the drug in front of the cornea was prolonged.	[96]
Natamycin	Eye inflammation	In vitro permeability was 3.3 times better than commercial formulations and 5.2 times better than suspensions.	[97]
Brimonidine tartrate	Glaucoma	The residence time of the drug in the cornea was significantly prolonged, and the intraocular pressure was significantly reduced.	[98]
**Ion-sensitive in situ gels**			
Ketotifen	Seasonal allergic conjunctivitis	The retention time of the drug in front of the cornea was prolonged.	[101]
Terbinafine hydrochloride	Fungal keratitis	The optimized in situ gel prolonged the mean residence time of the drug and enhanced ocular bioavailability.	[102]
Pefloxacin mesylate	Conjunctivitis and corneal ulcers	The drug was released in vitro for up to 12 h, and the best prescription had good stability and a shelf life of 2 years.	[105]
Phenylephrine, tropicamide	Mydriasis	Compared with normal eye drops, the intensity and duration of pupil dilatation in rabbits were increased by 4 to 8 times.	[106]
**Posterior segment**			
**p** **H-sensitive in situ gels**			
Bear bile	Retinitis pigmentosa and age-related macular degeneration	The optimum prescription was biocompatible and nonirritating and prolonged the corneal retention time of the drug by approximately 3 times.	[99]

### 4.4. Nanoparticle Eye Drop Delivery Systems

Nanoparticles (typically 10–1000 nm in size) can deliver drugs to the posterior segment of the eye by passive or active (ligand-mediated) targeting [107]. Nanoparticles can promote drug penetration into the ocular layer, prolong the residence time of eye drops and reduce toxicity. In recent years, many nanoparticles have begun replacing invasive and surgical interventions in the treatment of many diseases of the posterior segment of the eye that cause visual impairment and even blindness. The nanoparticles for targeting the posterior segment of the eye mainly include liposomes, niosomes, dendrimers, solid lipid nanoparticles, nanosuspensions, microemulsions, polymer nanoparticles, and nanomicelles (Figure 5) [108].

#### 4.4.1. Liposome Eye Drop Delivery Systems

Liposomes are spherical vesicles composed of phospholipid and steroid (cholesterol) bilayers [109]. Liposomes can be divided into monolayers or multilayers and have a variety of drug-carrying properties. Both hydrophilic and lipophilic drugs can be encapsulated in liposomes, with hydrophilic drugs encapsulated in the core and lipophilic drugs in the bilayer. In addition, liposomes are biodegradable, biocompatible carriers that enhance drug penetration by binding to the corneal surface [110].

Liposome eye drop delivery systems improve the absorption, distribution, metabolism, and excretion (ADME) of drugs and have been investigated for the treatment of anterior and posterior segment diseases of the eye such as glaucoma, diabetic macular edema, endophthalmitis, and uveitis [111]. A recent study concluded that ophthalmic liposomes loaded with ganciclovir (GCV) prepared using reverse-phase evaporation resulted in 3.9 times higher corneal permeability of GCV than the corresponding solution after topical administration [112]. In addition, liposomes loaded with diclofenac increased the cumulative concentration of diclofenac in the retinal choroid by 1.8 times compared with diclofenac solution [113].

Generally, liposomes smaller than 200 nm are considered ideal because of maximum absorption, which decreases with increasing size, while liposomes with sizes of 600 nm show negligible absorption in the posterior segment of the eye [114]. In addition, positively charged liposomes can be noncovalently bonded to the negative charge carried by the mucosal layer, so positively charged liposomes remain in front of the cornea longer than negatively charged or neutral liposomes [115]. After 2.5 h of topical administration to rabbit eyes, ACV-loaded liposomes prepared using stearylamine as a cationic inducer were more concentrated in corneal tissue than anionic liposomes prepared using diacetyl phosphate as an anionic inducer or ACV solutions, indicating that the cationic liposomes were more readily absorbed into the whole corneal tissue [116]. Similarly, cationic liposomes loaded with ibuprofen significantly prolonged the time to peak (T_max_) to 100 min and the AUC to 1.53 times that of ibuprofen eye drops [117].

Furthermore, different approaches have been proposed to increase the residence time of liposomes in the precorneal area, such as the use of viscosity enhancers, in situ gelling polymers, and nanoparticle surface modifiers. The main purpose of surface modifiers is to maximize the interaction of the liposomes with eye structures and/or promote nanoparticle penetration [118,119]. Hyaluronic acid (HA)-modified liposomes loaded with doxorubicin (DOX) had longer corneal retention time than unmodified liposomes and free DOX [120]. Similarly, CS-modified liposomes loaded with TM showed a 3.18-fold increase in apparent permeability coefficient (P_app_) compared with commercial ophthalmic solutions [121]. In comparison with methazolamide (MTA) solution, MTA liposomal gel showed a significant reduction in IOP. The areas under the percentage decrease in IOP vs. time (h) curves (AUC_0–8 h_) were found to be 58 ± 0.03, 174 ± 0.04, and 222 ± 0.03 h^−1^ for MTA drug, MTA liposomes, and MTA liposomal gel. respectively. MTA liposomal gel, as opposed to traditional eye drops, maybe a suitable delivery medium for ocular distribution [122].

Transporter-targeted liposomes represent a new development in topical administration that has attracted a great deal of attention in recent years. Many membrane transporters. such as peptides, amino acids, glucose, lactate, and nucleosides/bases, have been identified in various ocular tissues, including the cornea, conjunctiva, and retina. Liposomes that target these transporters could significantly enhance drug penetration [15]. iRGD can be cleaved by proteases when specifically bound to integrin *αvβ*_3_ expressed by corneal epithelial cells. This cleaved iRGD can bind to neuroproteinase-1 (NRP-1), thereby activating the endocytic transport pathway [123]. Studies have shown that iRGD-modified liposomes could be used as an effective ocular drug delivery strategy. iRGD-modified liposomes loaded with brinzolamide effectively penetrated the corneal barrier after topical administration in an iRGD receptor-mediated manner, exhibiting a stronger and longer-lasting way therapeutic effect in glaucoma compared with commercially available brinzolamide eye drops [124]. Table 4 [112,113,125,126,127] lists liposome eye drop delivery systems in the application of the anterior and posterior segments of the eye.

#### 4.4.2. Niosome Eye Drop Delivery Systems

Niosomes are a special type of vesicle formed by an amphiphilic nonionic surfactant and cholesterol [128]. As an ocular drug delivery system, niosomes are biocompatible, biodegradable, structurally flexible, and suitable for loading hydrophobic and hydrophilic drugs [129]. As the nonionic surfactant replaces phospholipids, which are prone to oxidative degradation, niosomes are more stable and less prone to leakage than liposomes. In addition, because of the surfactant’s ability to open tight junctions and promote corneal permeability, niosomes also have higher ocular bioavailability than liposomes [130].

The ability of niosomes to enhance drug bioavailability and efficacy has attracted researchers to apply niosomal eye drop delivery systems for the treatment of diseases in the anterior and posterior segments [131,132]. Abdelbary et al. encapsulated gentamicin sulfate into niosomes to maintain and control the release of the drug. In vitro drug release results showed that gentamicin was released more slowly in niosomes than in the corresponding solutions. Hydrophilic–lipophilic balance (HLB) values also play an important role, with higher encapsulation efficiency and stability obtained by using surfactants with low HLB [133]. Pharmacodynamic studies showed that carbomer-coated niosomes loaded with acetazolamide reduced IOP by 33% and that the hypotensive effect lasted for 6 h after drip administration, four times as long as that of doxorubicin solution, which lasted about 1.5 h [134].

Niosomes coated with bioadhesive materials (CS, HA) or combined with hydrogels are more suitable for ocular drug delivery [135]. Zeng et al. developed HA-coated niosomes for ocular delivery of tacrolimus (FK506). Precorneal retention results showed that HA-coated niosomes significantly prolonged the retention time of FK506 over those of uncoated niosomes and suspensions. Aqueous humor pharmacokinetic tests showed that the AUCs of HA-coated niosomes were 2.3 and 1.2 times greater than those of suspension and uncoated niosomes, respectively. The synergistic enhancement of FK506 corneal permeability by the hybrid delivery system was observed and confirmed by confocal laser scanning microscopy. HA-coated niosomes promote ocular delivery of FK506 in terms of mucosal adhesion, precorneal retention, aqueous humor pharmacokinetics, and transcorneal permeability and are a promising method for ocular targeted delivery of FK506 [136]. After the installation of niosome-loaded gels into the eyes of rabbits, the IOP was prolonged, and the relative bioavailability in normal and glaucomatous rabbits of betaxolol was significantly higher than that of commercially available eye drops [137]. Table 4 [133,134,136,138] lists niosome eye drop delivery systems in the application of the anterior and posterior segments of the eye.

#### 4.4.3. Solid Lipid Nanoparticle Eye Drop Delivery Systems

Solid lipid nanoparticles (SLNs) can be defined as solid lipid matrices in the nanometer size range accommodating drugs that are stabilized by one or more surfactants [139]. Compared with other colloidal carriers, SLNs not only have good biocompatibility and biodegradability and high permeability [140] but can be produced on a large scale by a high-pressure homogenization method. More importantly, solid state SLNs can prevent or reduce the degradation of lipophilic drugs compared with liquid liposomes. Nanostructured lipid carriers (NLCs), which are also solid nanoparticles without a crystalline structure, are able to hold more drug than SLNs.

In recent years, SLN eye drop delivery systems have gained increasing attention as drug carriers [141]. SLN loaded with KTZ showed 2.5- and 1.6-fold increases in bioavailability in the aqueous humor and vitreous, respectively, compared with KTZ suspensions [142]. Similarly, NLC loaded with *N*-palmitoyleth anolamine (PEA), significantly increased the amount of PEA in the ocular tissue compared with PEA suspension [143].

It is generally accepted that SLNs with smaller sizes are more suitable for topical administration than larger ones. This can be attributed to the fact that the small size facilitates faster travel of SLNs through the mucus layer of the tear film and favors corneal uptake by epithelial cells [144]. In addition, it has been reported that surface charge also affects nanoparticle uptake, with SLNs on cationic surfaces showing higher permeability than neutral or anionic surfaces [145].

Improving the adhesion of SLNs was found to be an effective strategy for improving the ocular bioavailability of drugs [146]. Liu et al. developed coumarin-6-loaded NLC (C6-NLC) using melt emulsion technology followed by surface modification with CS-hydrochloride (CH) and CS-NAC. The result of in vitro mucosal adhesion demonstrated that the presence of CS-NAC on the C6-NLC surface provided the most obvious enhancement in adhesion because of the formation of both noncovalent (ionic) and covalent (disulfide bridges) interactions with mucus chains. Furthermore, the higher the concentration of CS-NAC was, the longer the retention time of the nanoparticles in the ocular tissue was. In addition, corneal penetration results showed that CS-NAC-NLC particles were able to penetrate the entire corneal epithelium, mainly via a transcellular pathway [147]. SLNs loaded with cyclosporine were prepared by Battaglia et al. using a coalescence method, and nanoparticles with different surface charges were obtained by using different stabilizers. After cyclosporine was labeled with fluorescent probes, the interaction of SLNs with the cornea was studied by fluorescence microscopy, and the accumulation and penetration of the drug in the cornea was calculated. The results showed that cationic (CS-coated) SLNs enhanced drug accumulation and penetration in rabbit corneas compared with anionic and nonionic SLNs and other reference formulations. The reason for this phenomenon may be the increased interaction of CS with the nanoparticles and the corneal epithelium [148].

Furthermore, targeted SLNs are an effective strategy for improving the ocular bioavailability of drugs. Using Wistar rats, Delgado et al. measured the in vivo transfection efficiency of SLN conjugated with protamine, dextran, and two plasmids coding for an enhanced green fluorescent protein (EGFP). The SLN formulation substantially increased the expression of EGFP in the cornea for topical administration. This success was attributed to the careful selection of nanoparticle components, especially the presence of dextran in SLN protamine–DNA complexes, which allowed targeted internalization through clathrin-mediated endocytosis, something fundamental for the uptake of protamine–DNA conjugates [149]. Table 4 [141,142,150,151,152,153] lists lipid nanoparticle eye drop delivery systems in the application of the anterior and posterior segments of the eye.

#### 4.4.4. Polymer Nanoparticle Eye Drop Delivery Systems

Polymeric nanoparticles (PNPs) are structures with diameters ranging from 10 to 1000 nm. Depending on their structure, polymeric nanoparticles can be divided into nanocapsules and nanospheres. The former encapsulate drugs inside the formed polymer lattice, while the latter consist of homogeneous dispersions of drugs into the polymer lattice. For ocular applications, the advantages of PNPs as a drug delivery system include high stability, controlled drug release, specific tissue targeting, and high retention times due to the adherent polymeric material [154].

The polymers used in nanoparticles are natural materials or modified polymers, such as the natural materials gelatin, CS, dextran sulfate, and hyaluronic acid [155] and the synthetic polymers PLA [156], poly(lactic acid-hydroxyacetic acid) (PLGA) [157], poly(ε-caprolactone), polyacrylamide, and polyacrylate. PNPs can also be made of inorganic materials, such as silica [158]. CS and its derivatives are commonly used as materials for adherent nanoparticles because of their adhesion properties, antibacterial activity, and ability to facilitate drug penetration by opening tight intercellular junctions [159]. CS nanoparticles loaded with carteolol (CRT) were prepared by Ameeduzzafar et al. with a particle size of 243 nm, drug loading of 49.21 ± 2.73%, and entrapment efficiency of 69.57 ± 3.54%. In vitro release studies showed a sustained release for 24 h as compared with drug solution, and scintigraphy study demonstrated good spread and retention in the precorneal area as compared with the aqueous CRT solution and prolonged reduction in intraocular pressure [160]. Karava et al. found that pure CS and its derivatives with 2-acrylamido-2-methyl-1-propanesulfonic acid (AAMPS) and [2-(methacryloyloxy)ethyl]dimethyl-(3-sulfopropyl)ammonium hydroxide (MEDSP) could both be used as carriers for PNPs for intraocular delivery of dexamethasone sodium phosphate (DxP) and chloramphenicol (CHL) [161]. In addition to acting as a backbone material for PNPs, CS and its derivatives can also be coated on the surface of nanoparticles to impart adhesion properties to the nanoparticles. For example, PLA nanoparticles loaded with rapamycin had good retention on the corneal surface when coated with chitosan [162].

The small size of PNPs is a very good feature for their promising characteristics for diminished irritation in the corneal tissue and capacity to sustain the delivery of the drug with further avoidance of multiple administrations [163]. In addition, surface charge also has a certain effect on the bioavailability of PNPs. In general, the higher the surface charge is, the more favorable the drug delivery is. For example, melatonin-loaded nanoparticles prepared with PLGA-PEG successfully reduced IOP in rabbits, and the hypotensive effect was better than that of melatonin-loaded nanoparticles prepared with PLGA. The explanation for this phenomenon was that the increased zeta potential of PLGA-PEG nanoparticles relative to PLGA nanoparticles allowed for better charge interactions with the cornea, producing a longer hypotensive effect [164].

PNPs surface modified with antibodies, vitamins, peptides, and inducers have shown strong uptake in specific tissues. According to Kompella et al., surface-modified PNPs showed 64% and 74% higher transport of norepinephrine and transferrin, respectively, compared with non-surface-modified PNPs. This suggests that surface modification enables PNPs to rapidly and efficiently enter and/or cross the cornea and conjunctiva [165]. Sharma et al. investigated the efficiency and toxicity of 2 kDa polyethylene glycol with gold nanoparticles (PEI2-GNP) in the delivery of genes to human corneas (in vitro) and rabbit corneas (in vivo). The results showed that the hybrid nanoparticles could efficiently deliver genes to the human cornea without altering cell viability. Significant particle uptake was observed in the rabbits’ aqueous humor following topical administration of the hybrid nanoparticles, with gradual clearance over time. Furthermore, slit-lamp biomicroscopy of live animals following topical administration revealed no inflammation or erythema and only moderate cell death and immune response, suggesting the potential use of PEI2-GNP in corneal gene therapy [166]. Table 4 [161,167,168,169,170] lists polymer nanoparticle eye drop delivery systems in the application of the anterior and posterior segments of the eye.

#### 4.4.5. Micelle Eye Drop Delivery Systems

Micelles are nanocarriers in size and have external hydrophilic polar heads and internal hydrophobic fatty acyl chains, which allow them to deliver water-insoluble drugs and protective molecules [171]. In exceptional cases, amphiphilic copolymers or surfactants may be misaligned to form “antimicelles” for encapsulating hydrophobic drugs [172,173]. In addition, micellar delivery systems have the advantage of being easier to prepare and having relatively small particle size compared with other nanocarrier drug delivery systems [174].

Micelles are emerging as a novel platform for drug delivery to the anterior segment of the eye. A methoxy polyethylene glycol-hexyl-substituted poly(lactic acid) (MPEG-hexPLA) micelle was used for the ocular delivery of cyclosporine A (CsA). The MPEG-hexPLA micelle formulation was well tolerated in the eye and represented a promising drug carrier for the treatment of eye diseases involving cytokine activation, such as dry eye syndrome and autoimmune uveitis, or for the prevention of corneal transplant rejection [175]. Zhang et al. developed a micellar eye drop using dipotassium glycyrrhizinate (DG) as a carrier and hesperidin (Hes) as a model drug. The optimized DG-Hes had a mean micelle size of 70.93 ± 3.41 nm and a polydispersity index of 0.11 ± 0.02. DG-Hes significantly improved the passive penetration, corneal penetration, and ocular bioavailability of Hes. In vitro antimicrobial activity tests showed that the lowest inhibitory and lowest bactericidal concentrations of DG-Hes ophthalmic solution were lower than those of free Hes. DG-Hes ophthalmic solution also significantly reduced the symptoms of ocular infection in a rabbit model of bacterial keratitis compared with Hes suspension. These results suggest that DG-Hes eye drops may be useful as a new ophthalmic agent for the treatment of ocular diseases, particularly bacterial eye diseases [176].

In addition, because of their small particle size, more and more micelles can deliver drugs to the fundus [177]. A nanomicellar system using vitamin E tocopherol polyethylene glycol succinate (TPGS) (Vit E TPGS) and octoxynol-40 (Oc-40) as polymeric substrates and rapamycin as a model drug resulted in very high rapamycin concentrations (362.35 ± 56.17 ng/g) in the retinal tissue of rabbit eyes after topical drops. No drug was found in the vitreous, indicating that rapamycin was sequestered in the lipid-like retinal tissue [178]. In addition, [179] showed that the tissue concentrations of DEX in the sclera, retinal choroid, and vitreous humor after multiple topical infusions of dexamethasone nanomicelles were 112.75 ± 53.09 ng/g, 67.32 ± 26.49 ng/g, and 3.85 ± 1.75 ng/g, respectively. This suggests that micelles may deliver the drug to the posterior part of the eye during topical administration by diffusion through the conjunctival–scleral route.

Positively charged micelles can improve the bioavailability of drugs. A micellar system consisting of the polyoxygenated nonionic surfactant pluronic1 F127 (F127) and the cationic polyelectrolyte chitosan (CH) loaded with DEX resulted in a significant increase in the in vitro release and transport of DEX in Caco-2 cell monolayers compared with the F127 micellar system without CH. Pharmacokinetic results in rabbit eyes showed 1.7-fold and 2.4-fold increases in the bioavailability of the F127 and F127/CH micelle systems, respectively, compared with the standard DEX suspension. These suggested that micellar systems improved the intraocular absorption of DEX [180].

In addition, micellar eye drops with active targeting can deliver as much drug to the posterior end of the eye. A cyclic cidofovir transporter-targeting lipid prodrug (B-C12-CDF) micelle delivery system prepared by linking the lipid chain (C-12) and the targeting portion (biotin) to cidofovir and its analogues (CDF) could be targeted to retinal tissue after topical administration. Better retinal targeting can be achieved by biotin-specific targeting of a sodium-dependent vitamin transporter that is highly expressed in the retina. Stable drug delivery to the retina and vitreous layer is possible if the drug is loaded into a polymeric carrier that controls prodrug release [181]. Table 4 [175,179,180] lists micelle eye drop delivery systems in the application of the anterior and posterior segments of the eye.

#### 4.4.6. Nanosuspension Eye Drop Delivery Systems

Nanosuspensions are two-phase colloidal dispersions systems in which drug particles are dispersed into an aqueous medium. Nanosuspensions provide an important and useful method for improving the bioavailability of low water-soluble drugs by reducing the particle size of the drug to the submicron range and stabilizing the drug with polymers, surfactants, or a mixture of both [182,183,184]. Because of the large surface area of the nanoparticles, the rate of drug release may be sufficient to maintain the effective drug concentration in the tear film, resulting in significant bioavailability. In addition, nanosuspensions also have the advantages of low irritation, required reduced dose, prolonged drug release, reduced systemic toxicity of the drug, and prolonged residence time of nanoparticles on the corneal surface [185].

Nanosuspensions make it possible to deliver hydrophobic drugs to the eye. For example, in a recently published report, Kassem et al. prepared and evaluated nanosuspension formulations of prednisolone, hydrocortisone, and dexamethasone for topical ocular delivery. In vivo tissue distribution studies of the glucocorticoid nanosuspensions demonstrated significantly higher levels in anterior chamber tissues relative to those of solutions and microcrystalline suspensions of similar compounds [186].

Cationic surfactants are often chosen as stabilizers for nanosuspensions because their positive charge improves the residence time of the drug in front of the cornea through noncovalent binding to negatively charged mucins [187]. Flurbiprofen (FLU) polymer nanoparticle suspensions prepared using eudragit RS100s and RL100s as polymer resins were effective in preventing severe myasthenia gravis caused by extracapsular cataract surgery compared with conventional eye drops [188]. Shi et al. developed a cationic nanosuspension of CS and methoxy polyethylene glycol-poly(ε-caprolactone) (MPEG-PCL) for ocular delivery of diclofenac (DIC). In vivo pharmacokinetic studies showed enhanced retention and permeability, higher concentration in aqueous humor (C_max_), and better bioavailability of the DIC/MPEG-PCL-CS nanosuspension compared with commercial DIC ophthalmic solutions [189]. Table 4 [188,190] lists nanosuspension eye drop delivery systems in the application of the anterior and posterior segments of the eye.

#### 4.4.7. Microemulsion Eye Drop Delivery Systems

Microemulsions (MEs) are thermodynamically stable systems consisting of water, oil, and surfactants/cosurfactants [191]. Depending on the dispersed phase and the dispersion medium, emulsions can be classified as oil-in-water emulsions (O/W), water-in-oil emulsions (W/O), or composite emulsions [192]. As a drug reservoir, MEs can deliver both hydrophilic and hydrophobic drugs to the cornea [193]. In addition to this, they have a wide range of properties: natural biodegradability, nanoscale droplet size, sterility, good solubility in either the innermost oil phase or the oil–water interface, and good ocular absorption [194]. Nanoemulsions (NEs) are also nanostructured emulsion consisting of an aqueous phase, an oil phase, a surfactant, and a cosurfactant and differ most from MEs in that they have a smaller particle size.

Researchers successfully developed some ME eye drop delivery systems that can be used in ophthalmic applications [195]. The optimized MEs had good stability and good adhesion and penetration to the corneal surface. The concentration of gatifloxacin was increased twofold over that of the conventional dosage form [196]. Compared with conventional DEX formulations, MEs prepared by titration penetrated more easily into the immediate anterior segment, with longer drug release times and higher bioavailability [197].

Changing the surface charge of MEs or surface modification is a common method to improve ocular bioavailability [198]. Ying et al. prepared a series of MEs using C6 as a fluorescent marker to explore the effect of different types of MEs on drug delivery to the posterior segment of the eye. Fluorescence analysis of eyes collected 30 min after topical infusion of MEs showed that surface-modified MEs containing CS and polyoxyethylene ether 407 resulted in increased fluorescence intensity in the retina compared with unmodified MEs. This phenomenon may be caused by the MEs through electrostatic interaction with the ocular cell membrane or by increasing the retention time on the ocular surface. Alternatively, MEs may enter the retina via a noncorneal pathway [199].

Incorporating MEs into in situ gels is also an effective strategy to improve ocular bioavailability [200]. Gan et al. prepared a CsA-loaded ME using castor oil, solutol HS 15 (surfactant), glycerol, and water, which were then dispersed in Kelcogel^®^ solution to form an in situ electrolyte-triggered ME gelling system. In vitro, the viscosity of the CsA ME Kelcogel^®^ system increased dramatically when diluted with artificial tears and exhibited pseudoplastic rheology. In vivo results showed that the AUC_(0–32 h)_ of corneal CsA with the ME Kelcogel^®^ system was approximately three times greater than that of the CsA ME [201]. Tajika et al. studied and evaluated the effectiveness of the terbinafine hydrochloride NE in situ gel system for the treatment of fungal keratitis. Under in vivo histopathological assessment, the NE in situ gel system showed no irritation in the studied ocular tissues (cornea, iris, retina, and sclera) compared with the control group [202]. Table 4 [195,196,203,204] lists emulsion eye drop delivery systems in the application of the anterior and posterior segments of the eye.

#### 4.4.8. Dendritic Polymer Eye Drop Delivery Systems

Dendritic polymers have been used extensively for drug delivery since the 19th century, and in recent years, their use in eye diseases has attracted the attention of scientists worldwide. Dendritic polymers are spherical, tree-branched, nanostructured polymers consisting of a central molecule called the “core” and side-chain molecules called the “branches” [205]. Dendritic polymer nanoparticles are typically smaller than 100 nm in size, and the peripheral functional groups (neutral, negative, or positive) can receive secondary surface modifications and are therefore endowed with a range of excellent properties for ophthalmic drug delivery, such as prolonged corneal residence time and enhanced permeability [206].

Poly(amidomine) (PAMAM) is the most studied, widely characterized, and commercialized dendrimer species for drug and gene delivery [207]. However, some PAMAM dendritic macromolecules are toxic in cells and animals because of their polycationic properties [208]. It has been shown that modifying the amino groups at the periphery of dendrimers with polyethylene glycol chains can reduce their toxicity and increase their biocompatibility [209,210]. PAMAM dendrimers with -OH or -COOH terminal molecules did not induce cytotoxicity and could have their low generation removed intact by urine [211]. PAMAM dendrimer eye drop delivery systems are also becoming more widely used. Vandamme and Brobeck prepared several series of PAMAM dendrimers using pilocarpine nitrate and tropicamide as model drugs. Using New Zealand albino rabbits as an in vivo model, the ocular tolerance and ocular retention time of PAMAM dendrimer solutions were assessed qualitatively and quantitatively after a single drop. The average ocular residence time of aqueous PAMAM dendrimer solutions (generations 1.5, 2, 3.5, and 4) was comparable to that of 0.2% *w*/*v* carbomer solutions under the same experimental conditions, and the ocular residence time of dendrimer (generation 2) solutions was even significantly longer than that of carbomer or HPMC solutions [212]. In addition, puerarin PAMAM dendrimer complexes showed a prolonged corneal residence time and 1.3-fold and 2.0-fold increases in the main pharmacokinetic parameters C_max_ and AUC, respectively, compared with puerarin solution [213].

Dendritic molecules also show a more significant advantage in delivering drugs to the posterior segment of the eye. Yavuz et al. constructed various anionic DEX–PAMAM complexes and evaluated the effect of DEX delivery to the posterior eye. When DEX–PAMAM (generations 4.5) complexes were administered topically, in vivo tissue distribution studies in rabbit eyes showed that the highest drug levels in the vitreous, retina–choroid, and sclera were 125.4 ± 90.0 μg/g, 329.8 ± 122.8 μg/g, and 1150.5 ± 232.9 μg/g, respectively, and that these concentrations were higher than those of DEX suspensions by 8.7, 8.9, and 3.99 times, respectively. In addition, in vitro transport studies of various DEX–PAMAM complexes in the rabbit cornea and sclera–retina were investigated and showed that DEX penetrated through the cornea and sclera–retina into the posterior eye [214]. Liu et al. designed a gene delivery system for osmolyte based on electrostatic binding to target the retina via a noninvasive delivery route. It was prepared with red fluorescent protein pellets (pRFP) and/or a low molecular weight polyamide dendrimer (G3 PAMAM). After installation into the conjunctival sac of rats, the intact complex rapidly penetrated from the ocular surface to the fundus and remained in the retina for more than 8 h, which resulted in efficient expression of RFP in the posterior segment. The intraocular distribution of the complexes suggested that the plasmids were absorbed into the eye via a noncorneal pathway [215]. Table 4 [212,216] lists dendritic polymer eye drop delivery systems in the application of the anterior and posterior segments of the eye.

**Table 4 pharmaceutics-14-01150-t004:** Nanoparticle eye drop delivery systems in the application of the anterior and posterior segments of the eye.

Model Drugs	Indications	Main Findings	Ref.
**Anterior segment**			
**Liposomes**			
Ganciclovir	Eye infections	The AUC of the aqueous humor concentration–time profile of ganciclovir liposomes was found to be 1.7 times higher than that of ganciclovir solution.	[112]
Timolol maleate	Glaucoma	The P_app_ and J_ss_ of timolol maleate liposomes were 1.50 times higher than that of the commercialized eye drops.	[125]
Azithromycin	Dry eye	Liposomes enhanced corneal permeation approximately twofold over that of azithromycin solution.	[126]
**Niosomes**			
Gentamicin	Eye infections	Niosome had a slower release rate than gentamicin sulphate compositions.	[133]
Acetazolamide	Glaucoma	Niosome had higher ocular bioavailability than drug solution.	[134]
Tacrolimus	Corneal allograft rejection	The AUC of niosomes was 2.3 times greater than that of suspension.	[136]
Latanoprost	Glaucoma	The reduced IOP of niosomes was significantly longer than commercial eye drops.	[138]
**Solid lipid nanoparticles**			
Ketoconazole	Ophthalmic mycoses	SLNs had higher ocular bioavailability than ketoconazole suspension.	[141]
Methazolamide	Glaucoma	SLNs had higher therapeutic efficacy, later occurrence of maximum action, and more prolonged effect than drug solutions and commercial products.	[150]
Methazolamide	Glaucoma	SLNs showed a significantly prolonged decreasing intraocular pressure effect compared with methazolamide solution.	[151]
**Polymer nanoparticles**			
Dexibuprofen	Eye inflammations	NPs were confirmed to be more effective to treat and prevent ocular inflammation than dexibuprofen solution.	[161]
5-fluorouracil	Squamous cell carcinoma	5-FU level in the aqueous humor of the rabbit eye was significantly higher than that due to 5-FU solution.	[167]
Daptomycin	Bacterial endophthalmitis	The antimicrobial activity of daptomycin was preserved when the antibiotic was encapsulated into NPs.	[168]
Pranoprofen	Eye inflammations	The corneal permeation coefficient of NPs was four times higher than that of commercial eye drop formulations and freeform drug solutions groups.	[169]
Fluocinolone acetonide	Uveitis, posterior uveitis, and panuveitis	NP eye drops showed greatly prolonged residence time of the drug on the ocular surface.	[170]
**Micelles**			
Cyclosporin A	Eye inflammations	The micelle formulation was well tolerated in the eye and represented a promising drug carrier for the treatment of eye diseases.	[175]
**Nanosuspensions**			
Flurbiprofen	Cataract	Drug levels in the aqueous humor were higher after the application of the nanosuspensions.	[188]
Hydrocortisone	Inflammation disorders of the eye	The AUC was significantly higher than that of the hydrocortisone solution.	[190]
**Microemulsions**			
Timolol maleate	Glaucoma	MEs had higher drug-loading and transport rates than control.	[195]
Gatifloxacin	Bacterial keratitis	MEs had good stability, greater corneal adherence, and permeability.	[196]
Dexamethasone	Uveitis	An improved therapeutic effect occurred for the treatment of uveitis.	[203]
Sirolimus	Immunosuppressants	Suitable for the immunomodulatory treatment of ocular surface disorders.	[204]
**Dendritic polymers**			
Pilocarpine nitrate and tropicamide	Albino	The ocular residence time of dendrimer (generation 2) solutions was significantly longer even than that of carbomer or HPMC solutions.	[212]
Gatidloxacin	Eye inflammations	Enhanced corneal transport and increased antimicrobial activity.	[216]
**Posterior segment**			
**L** **iposomes**			
Diclofenac	Macular edema	Liposomes prolonged the retention time in the cornea and allowed higher bioavailability of diclofenac sodium.	[113]
Triamcinolone acetonide	Pseudophakic cystoid macular edema	The best corrected visual acuity and central eye socket thickness in the patient improved significantly.	[127]
**Solid lipid nanoparticles**			
N-palmitoyleth anolamide	Retinal inflammation	NLCs significantly increased the levels of PEA in the vitreous and retina compared with a drug suspension.	[142]
Atorvastatin	Age-related macular degeneration	SLNs were 8 and 12 times more bioavailable in the aqueous and vitreous humor, respectively, than free atorvastatin.	[152]
Triamcinolone acetonide	Macular edema	NLCs could deliver lipophilic active substances to the posterior segment of the eye via both corneal and noncorneal pathways.	[153]
**Micelles**			
Dexamethasone	Posterior uveitis	Micelles delivered the drug to the posterior part of the eye, probably by diffusion through the conjunctival–scleral pathway.	[179]
Dexamethasone	Diabetic macular edema	The AUC values showed 1.7- and 2.4-fold increases in bioavailability with Pluronic1 F127 and Pluronic1 F127/chitosan micelle systems, respectively, as compared with a standard dexamethasone suspension.	[180]

Notes: AUC, area under the curve; P_app_, apparent permeability coefficient; J_ss_, the flow rates of steady-state; IOP, intraocular pressure; SLNs, solid lipid nanoparticles; NPs, nanoparticles; 5-FU, 5-fluorouracil; MEs, microemulsions; HPMC, hydroxypropyl methyl cellulose; PEA, *N*-palmitoyleth anolamide; NLCs, nanostructured lipid carriers.

## 5. Novel Eye Drop Products and Clinical Trials

Prodrug-based designs of eye drops have proven successful. Prostaglandin F2α analogues such as bimatoprost (Lumigan^®^), travoprost (Travatan^®^), and latanoprost (Xalatan^®^) are used to lower IOP in glaucoma. Loteprednol etabonate (Lotemax^®^), a topical corticosteroid based on retrometabolic drug design, has also proven successful in the treatment of ocular allergies and inflammation. Some marketed eye drop products have also been developed based on in situ gel delivery systems. For example, Tiopex^®^ is an ophthalmic gel eye drop consisting of the pH-sensitive in situ gel polymer carbopol 974P for the delivery of TM for the treatment of glaucoma. Similarly, Pilopine HS^®^, an aqueous gel containing pilocarpine hydrochloride formed from carbopol 940, is used for the control of intraocular pressure. In addition, Timoptic^®^ GFS and Timoptic-XE are ion-sensitive in situ gels that have been approved for the treatment of glaucoma.

A number of novel eye drops are being studied in related clinical trials. Oculis, an ophthalmic pharmaceutical company, has developed a drug delivery platform based on cyclodextrin nanoparticles. Animal tests and preliminary clinical trials have shown that this technology has the potential to increase drug concentrations in ocular tissues, including the retina, for the treatment of retinal diseases such as DME. This study is now being studied in a clinical phase 2 trial (NCT05343156). A multicenter, two-arm randomized clinical trial investigating and comparing the clinical efficacy of 3% chloroprocaine gel and 0.5% tetracaine ophthalmic solution as local anesthetics in emulsification surgery will be conducted in approximately four European countries (NCT04685538). A study on the effect of lipid-based eye drops on tear film lipid layer thickness is being conducted in a clinical trial (NCT03399292). The clothes lane study on brimonidine tartrate NE eye drops in patients with ocular graft versus host disease (OGVHD) is now being studied in a clinical phase 3 trial. In addition, a randomized, placebo-controlled, double-masked, multicenter phase 3 study of brimonidine tartrate NE drops for the treatment of dry eye disease (DED) will be conducted at approximately 25 centers in the United States. (NCT03785340).

## 6. Conclusions

Delivering drugs efficiently and harmlessly into lesions in the posterior segment of the eye is a big challenging task. As a noninvasive delivery system, topical administration is the first treatment option for many ocular diseases because of the potential to eliminate the risks of intraocular injection and the toxicity of systemic drug delivery. The bioavailability of conventional eye drops is low because of the presence of multiple dynamic and static barriers, such as tear drainage, corneal barriers, and conjunctival elimination. To overcome these limitations, many pharmacists have conducted intensive research into topical ophthalmic delivery systems in recent decades to produce safe and effective eye drops with greater efficacy and greater safety. Some novel eye drop delivery systems, including prodrugs, cyclodextrins, in situ gels, and nanoparticles, are now being used to treat a variety of eye diseases. These novel eye drop delivery systems could increase the local concentration of the drug in the eye by increasing the retention time in front of the cornea and facilitating penetration. At the same time, they also allow more drugs to be delivered to the tissues at the posterior segment of the eye via the corneal route or the conjunctival–scleral route, which avoids the need for invasive surgery. These novel eye drop delivery systems have shown improved efficacy in the treatment of anterior and posterior segment disorders of the eye. Nevertheless, the negative phenomena associated with them should be taken into account. Prodrugs may lead to increased lipid solubility and toxicity of the drug. Highly viscous in situ gels may lead to blurred vision. For some stimulus-sensitive materials with relatively weak gelation efficiency, the use of high concentrations of materials or combinations of several materials can increase their toxicity. Nanoparticles may suffer from low encapsulation rates, poor stability, and a tendency for the drug to leak during storage. Therefore, not only the prescribing factors but the stability and sterility of the product need to be considered in the early stages of eye drop development. Further research and development of methods for improving drug preparation and storage are also needed to ensure the efficacy and safety of eye drops. As research into ophthalmic drug delivery systems continues, new safe and effective eye drops are expected to replace or surpass current invasive techniques in the near future and are expected to be used to treat a wide range of ocular diseases.

## Figures and Tables

**Figure 1 pharmaceutics-14-01150-f001:**
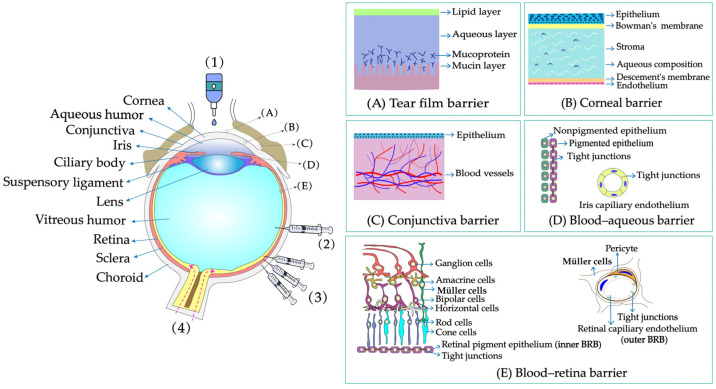
Structures of the eye, routes of drug delivery to the eye, and ocular barriers to drug delivery. The structures of the eye from the outside to the inside are the conjunctiva, cornea, aqueous humor, iris, ciliary body, lens, suspensory ligament, vitreous humor, sclera, choroid, and retina. Routes of ocular administration include (1) topical administration; (2) vitreous injection; (3) periocular injection; and (4) systemic administration. The main barriers to ocular administration consist of (**A**) the tear film barrier: composed of lipid, aqueous, and mucin layers. Acts as a defensive barrier against the entry of foreign objects into the cornea and conjunctiva; (**B**) the corneal barrier: consists mainly of endothelium containing tightly connected epithelial cells, water-soluble stroma, and a single layer of endothelial cells. Acts as a barrier to prevent the absorption of drugs from the tear fluid into the anterior chamber after topical administration; (**C**) the conjunctival barrier: a mucous membrane consisting of the conjunctival epithelium and underlying vascular connective tissue. Absorption area is larger than that of the cornea, and drugs are easily absorbed into the body circulation through capillaries; (**D**) the blood–aqueous barrier (BAB): located in the anterior segment of the eye. Formed by the iris capillary endothelium and the nonpigmented epithelium of the ciliary body, both of which contain tight junctions. Prevents the passage of drugs from the blood (systemic) into the aqueous humor; and (**E**) the blood–retinal barrier (BRB): located in the posterior segment of the eye. Formed by the retinal pigment epithelium (outer BRB) and the endothelial membrane of the retinal blood vessels (inner BRB), both of which contain tight junctions. The tight junctions restrict the entry of the drugs from the blood (systemic) into the retina/aqueous humor.

**Figure 2 pharmaceutics-14-01150-f002:**
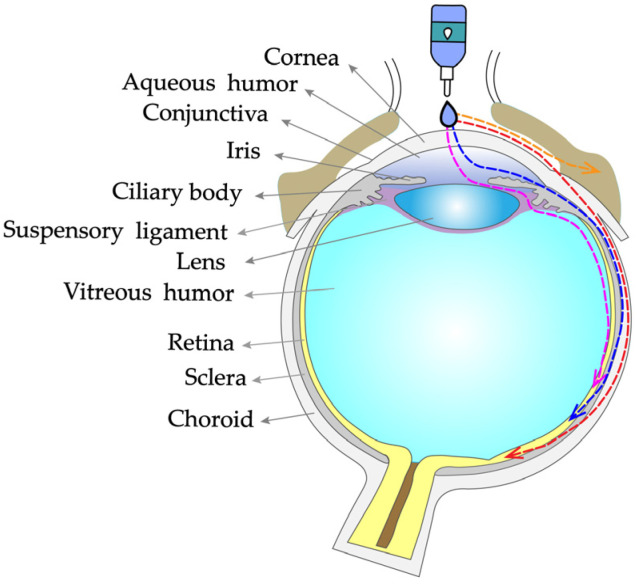
Different topical drug absorption routes from the cornea/conjunctiva–sclera to the posterior segment of the eye. Conjunctival–scleral route marked in red. Uveal–scleral route marked in blue. Transvitreal route marked in rose. Orange marks the systemic absorption route.

**Figure 3 pharmaceutics-14-01150-f003:**
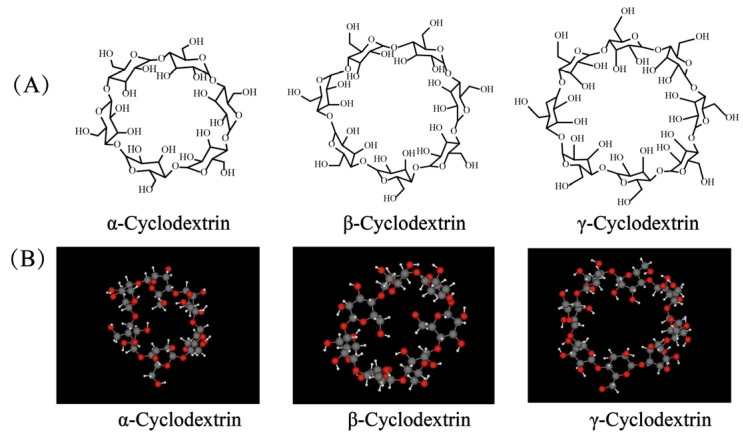
Structures of three common cyclodextrins: (**A**) planar structure; (**B**) 3D structure.

**Figure 4 pharmaceutics-14-01150-f004:**
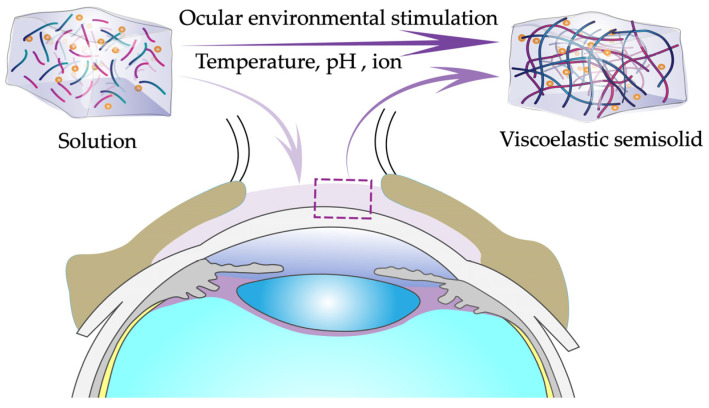
Diagram of in situ gel eye drops delivery systems in the eye.

**Figure 5 pharmaceutics-14-01150-f005:**
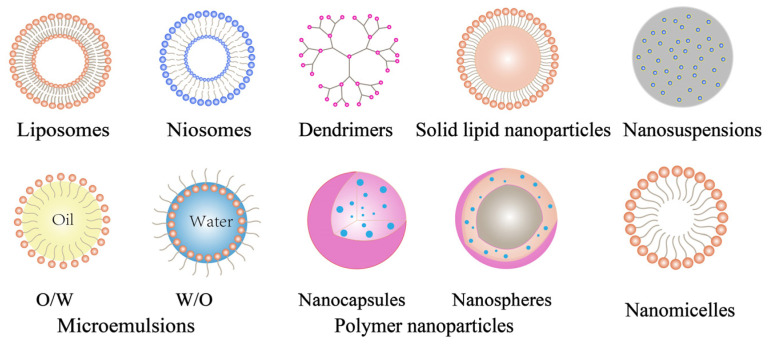
Schematic illustration of different nanoparticle eye drop delivery systems. The different nanoparticles include liposomes, niosomes, dendrimers, solid lipid nanoparticles, nanosuspensions, oil-in-water (O/W)- and water-in-oil (W/O)-type microemulsions, nanocapsule- and nanosphere-type polymer nanoparticles, and nanomicelles.

**Table 1 pharmaceutics-14-01150-t001:** Prodrug eye drop delivery systems in the application of the anterior and posterior segments of the eye.

Model Drugs	Prodrugs	Indications	Main Findings	Ref.
**Anterior segment**				
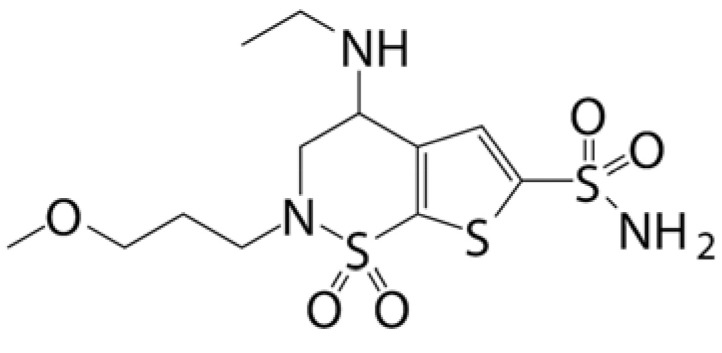 Brinzolamide	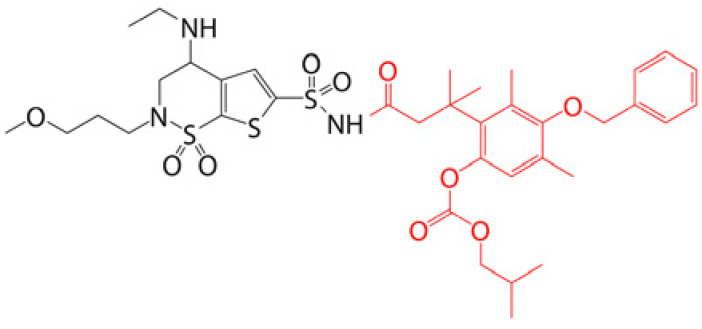 TML prodrug	Glaucoma	Prodrug penetrated corneal tissue more easily and was approximately five times more effective in reducing IOP than commercial eye drops.	[45]
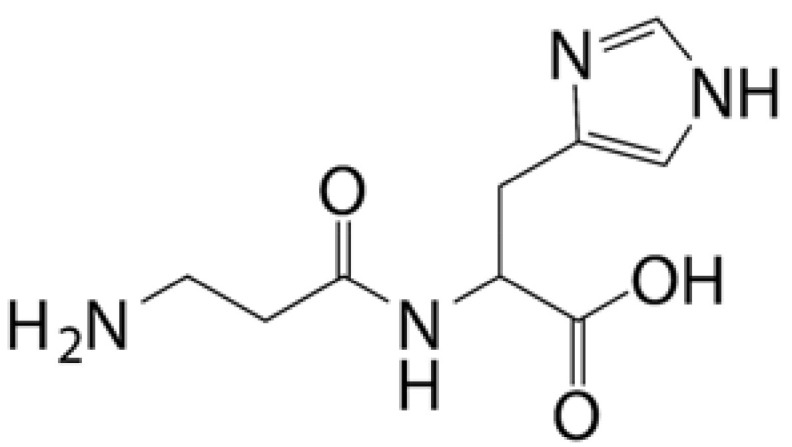 L-carnosine	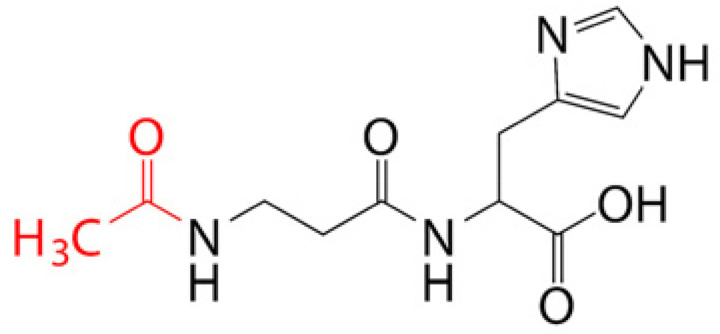 NAC	Cataracts	The intraocular retention time was prolonged, corneal permeability was increased, and bioavailability was significantly improved.	[46]
**Posterior segment**				
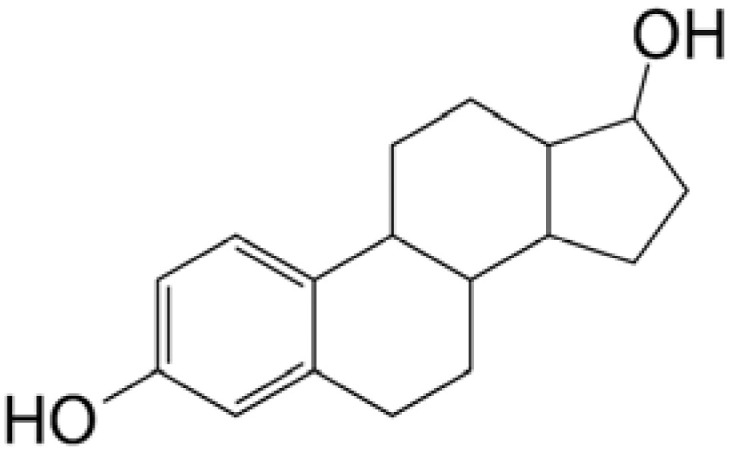 E2	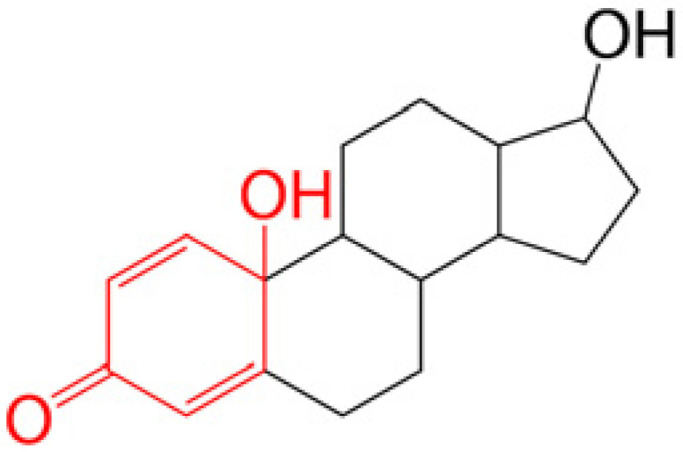 DHED	Retinal neuroprotection	Prodrug protected the retina more effectively and safely.	[52]
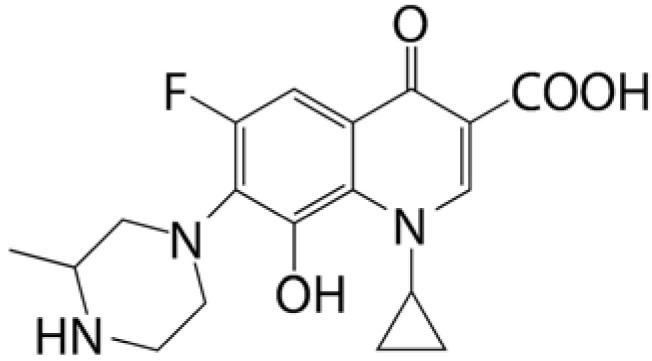 GFX	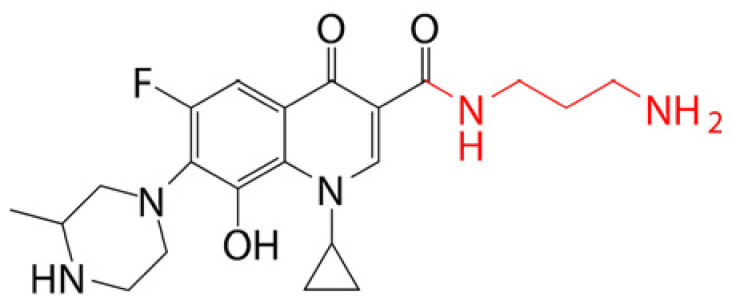 DMAP-GFX 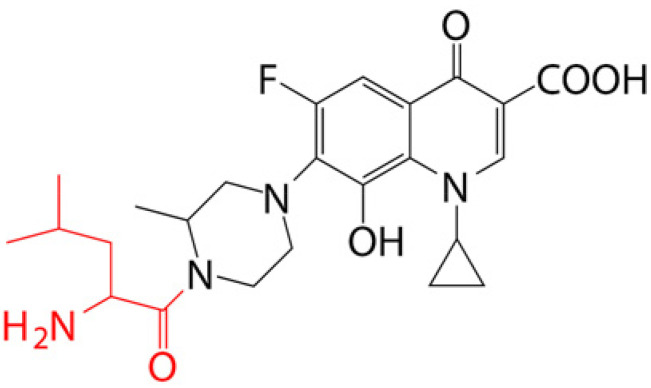 CP-GFX 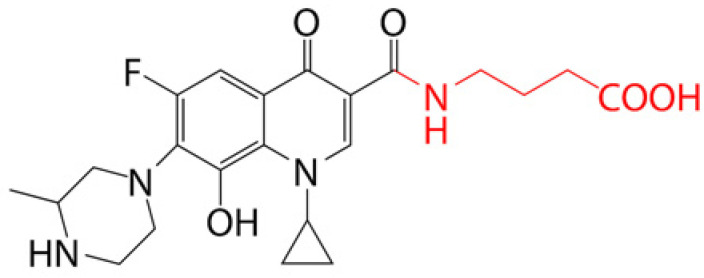 APM-GFX	Eye inflammation caused by the vitreous humor	The solubility and LogD of the drug were increased. Of the three prodrugs, DMAP-GFX could deliver the drug to the posterior part of the eye via OCT-mediated transport.	[53]

Notes: IOP, intraocular pressure; LogD, lipid–water distribution coefficient; DMAP-GFX, dimethylaminopropylgatifloxacin; OCT, organic cation transporter; E2, 17β-estradiol; DHED, 10β,17β-dihydroxyestra-1,4-dine-3-one; TML, trimethyl lock; NAC, *N*-acetylcarnosine; GFX, gatifloxacin; DMAP-GFX, dimethylaminopropylgatifloxacin; CP-GFX, carboxypropylgatifloxacin; APM-GFX, aminopropyl(2-methyl)-gatifloxacin. The red sites indicate the chemically modified functional groups of the active compounds.

**Table 2 pharmaceutics-14-01150-t002:** Cyclodextrin eye drop delivery systems in the application of the anterior and posterior segments of the eye.

Model Drugs	Indications	Main Findings	Ref.
**Anterior segment**			
Econazole nitrate	Eye infections	β-CD and HP-β-CD increased the solubility of EC by approximately threefold and fourfold, respectively.	[55]
Fluconazole	Eye infections	The retention time in front of the cornea was prolonged.	[56]
Latanoprost	Glaucoma	Stability and ocular bioavailability were higher than those of commercial eye drops.	[63]
Tacrolimus	Dry eye	Solubility was increased by approximately 42-fold.	[64]
Diclofenac sodium	Eye inflammations	Solubility was increased by approximately 20-fold.	[65]
**Posterior segment**			
Dexamethasone	Macular edema and branch retinal vein occlusion	More drugs were delivered to the retinal tissue.	[57]
Dexamethasone	Diabetic macular edema	The patient tolerated it well, with a reduction in central macular thickness and improved vision.	[66]
Celecoxib	Age-related macular degeneration and diabetic retinopathy	The amount of drug passing through semipermeable membranes, simulated vitreous, and sclera was increased.	[67]

Notes: β-CD, β-cyclodextrins; HP-β-CD, hydroxypropyl-β-cyclodextrin; EC, econazole nitrate; RPE, retinal pigment epithelium.

## Data Availability

Not applicable.

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
