# Peer review of "Novel Eye Drop Delivery Systems: Advance on Formulation Design Strategies Targeting Anterior and Posterior Segments of the Eye"

_pharmaceutics, 2022, doi:10.3390/pharmaceutics14061150_

Round 1
Reviewer 1 Report
There are many recently published reviews that deal with the same topic as this manuscript. For this reason I believe that a further review could be of interest only if a critical comment on them is made alongside the test of the selected articles. Instead in the present manuscript the authors briefly explain the content of the selected research papers, moreover sometimes citing other reviews rather than research articles. Furthermore, I find that even the choice of research articles to be cited has not been thought through enough. In fact, I see cited works that are sometimes not very significant while others that are much more significant are missing altogether. For all these reasons I believe that this review cannot be accepted for publication in this journal which has such a high impact factor.
Author Response
Response to Reviewer 1 Comments
Manuscript ID: pharmaceutics-1716667.
Title: Novel eye drop delivery systems: advance on formulation de-sign strategies targeting anterior and posterior segments of the eye.
Dear Editors and Reviewers:
Thank you for your letter and for the reviewers' comments on our manuscript. We appreciate you taking the time to read and review it. The comments were valuable and helpful in revising and improving our paper. We have carefully studied these comments and have made corrections in the manuscript using a revision model and the corrections are marked in red, which we hope you will approve. The main corrections and point-by-point responses to the comments in the thesis are set out below.
Responds to the reviewer’s comments:
Question: There are many recently published reviews that deal with the same topic as this manuscript. For this reason, I believe that a further review could be of interest only if a critical comment on them is made alongside the test of the selected articles. Instead in the present manuscript the authors briefly explain the content of the selected research papers, moreover sometimes citing other reviews rather than research articles. Furthermore, I find that even the choice of research articles to be cited has not been thought through enough. In fact, I see cited works that are sometimes not very significant while others that are much more significant are missing altogether. For all these reasons I believe that this review cannot be accepted for publication in this journal which has such a high impact factor.
Response: Thank you for your kind comments. We think that you must have read a lot of articles and reviews in the field of ocular drug delivery systems and have a comprehensive knowledge of the field. So, we think the comments you make are reasonable.
Indeed, novel ocular drug delivery systems are a hot topic in scientific research and have attracted the attention of many scholars, and as a result, many research articles or reviews have been submitted or published in recent years. As far as those published articles and reviews are concerned, I think that our review has the following advantages.
- Our review is broader in scope. In terms of reviews on topical delivery systems, Jumelle et al. reviewed the progress and limitations of in situ gel and nanoparticle eye drops (https://doi: 10.1016/j.jconrel.2020.01.057). Yellepeddi and Palakurthi reviewed the progress of prodrug and nanoparticle eye drops (https://doi: 10.1089/jop.2015.0047). Alvarez-Trabado et al. reviewed the design of lipid nanoparticles for topical delivery (https://doi: 10.1016/j.ijpharm.2017.09.017). In contrast to these reviews, our review includes studies on prodrug and cyclodextrin eye drops as well as in situ gel and nanoparticle eye drops. In addition, the previous reviews have mainly focused on the treatment of the anterior (https://doi: 10.1089/jop.2012.0200) or posterior segment of the eye (https://doi: 10.1080/17425247.2018; https://doi: 10.1016/j.ijpharm.2017.07.065). However, our review not only investigated novel eye drops delivery systems for the anterior segment but also addressed the posterior segment of the eye. In summary, our review on dosage forms and ocular diseases is broader in scope.
- Furthermore, our review is novel, because we not only presented the advantages and common materials used for prodrug, cyclodextrin, in situ gel, and nanoparticle eye drops, but we also investigated highly ocular bioavailability eye drops based on these novel delivery systems, such as self-assembled prodrug nanoparticles eye drops [49,50], cyclodextrin in situ gel eye drops [57,58], surface-modified nanoparticle eye drops [119,120,136,149,150,182] and ligand-targeted nanoparticle eye drops [123,150,167,183].
- More importantly, it is valuable and meaningful to study the application of novel eye drop delivery systems in the anterior and posterior segments of the eye. As far as the treatment of ocular diseases is concerned, it can improve the reference for non-invasive therapies. In terms of eye drop research, it can provide a reference for high ocular bioavailability eye drops.
We realized that some of the findings in my research were not expressed clearly enough in the manuscript. Therefore, we have revised this section. For example, dexamethasone (DEX)/γ-cyclodextrin (γ-CD) inclusion complexes reached concentrations of 29 ± 16 ng/g in the vitreous and 57 ± 22 ng/g in the retina of rabbits 2 h after topical administration [17]. In addition, they also found that both hydrophilic HP-β-CD and lipophilic RM-β-CD increased dexamethasone DEX concentrations after topical injection into the posterior segment of the eye [57]. The review in this part lacked not only a description of the concentrations of DEX/HP-β-CD and DEX/RM-β-CD after topical administration, but also a comparison of the data with DEX eye drops, so this part has been revised to Loftsson et al. found that the concentration of dexamethasone (DEX) in aqueous hu-mor was 66 ± 20 ng/g after 2 h of topical DEX ophthalmic solution. For 1.3% w/v DEX/HM-β-CD, the value was determined to be 320 ± 230 ng/g [55]. After a single ap-plication of 1.5% w/v DEX/γ-CD for 2 h, the concentration of DEX was 236 ± 67 ng/g in the aqueous humor, 29 ± 16 ng/g in the vitreous humor, and 57 ± 22 ng/g in the retina [17]. After 2 h of administration of 0.5 and 1.5% w/v DEX/ randomly methylated b-cyclodextrin (RM-β-CD) eye drops, the aqueous humor DEX levels were 1,190 ± 110 and 1,670 ± 630 ng/g, respectively. Levels in the retina were 33 ± 7 and 66 ± 49 ng/g, respectively, and in the optic nerve were 41 ± 12 and 130 ± 50 ng/g [55]. The above data suggested that γ-CD, hydrophilic HM-β-CD, and lipophilic RM-β-CD could enhance the local transport of DEX in the eye. Furthermore, the reason why lipophilic RM-β-CD resulted in higher DEX concentrations than HM-β-CD and γ-CD may be due to the fact that lipophilic RM-β-CD not only enhances drug delivery to the lipophilic cornea and sclera through the aqueous tear film but also reduces its barrier function by penetrat-ing the membrane (Line 263-276). In addition, more changes have been highlighted in blue in the revised edition.
There are some references in our review that were not very appropriate or were non-research articles. So, we have corrected these references, see the revised manuscript: [3], [9], [11], [14], [44], [60], [190].
Special thanks to you for your helpful comments!
We tried our best to improve the manuscript and made some changes in the manuscript. These changes which were marked in red font in the revised manuscript will not influence the content and frame work.
Thanks again for your kind help. We hope this revised version of our manuscript will be acceptable for publication. For any other questions, please feel free to contact us and we will response at our first time.
Sincerely yours,
Professor Chang-hong Wang
2022/5/15

Reviewer 2 Report
The manuscript entitled “Novel eye drop delivery systems: advance on formulation design strategies targeting anterior and posterior segments of the eye” reviews recent results and challenges on eye drop formulations. In my opinion, this is a comprehensive work written in a good language to please the potential readers. Nevertheless, I give some suggestions to improve it further:
- The abstract and introduction should be more focused. There are actually an overwhelmingly large number of reviews on ocular formulations. I recommend to emphasize better what this review adds to already available ones.
- Clinical application and commercial aspects are mentioned in the abstract but I feel that these are rather missing in most subsections and discussion is rather based on in vitro, ex vivo data. Please add some data where clinical trials or application can be found.
- Nanoparticles are explained very much in detail which reflects on the current interest in such systems. However, I recommend the authors to be more comparative than only descriptive. E.g., section 4.4.3 should start with the advantages OVER other systems rather definition and general descriptions.
- Please explain more in detail: “Niosomes are more stable than liposomes due to the replacement of phospholipids with non-ionic surfactants”.I cannot see the point why phospholipids are less stable than non-ionic surfactants.
- In 4.4.7, I have doubts if (all) micoemulsions are thermodynamically stable. Please clarify.
- I really miss the explanation of current challenges both in subsections and also in the conclusion. It looks like this is a finished work but probably we are at only the beginning of the development of highly efficient formulations.
Author Response
Dear Editors and Reviewers:
Thank you for your letter and for the reviewers' comments on our manuscript. We appreciate you taking the time to read and review it. The comments were valuable and helpful in revising and improving our paper. We have carefully studied these comments and have made corrections in the manuscript using a revision model and the corrections are marked in red, which we hope you will approve. The main corrections and point-by-point responses to the comments in the thesis are set out below.
Responds to the reviewer’s comments:
The manuscript entitled “Novel eye drop delivery systems: advance on formulation design strategies targeting anterior and posterior segments of the eye” reviews recent results and challenges on eye drop formulations. In my opinion, this is a comprehensive work written in a good language to please the potential readers. Nevertheless, I give some suggestions to improve it further:
Question1: The abstract and introduction should be more focused. There are actually an overwhelmingly large number of reviews on ocular formulations. I recommend to emphasize better what this review adds to already available ones.
Response: Thank you for your kindly comments. In recent years, many new advanced topical drug release systems have been developed to overcome the apparent limitations of traditional eye drops. For example, Jumelle et al. reviewed the advances and limitations of in situ gel and nanoparticle eye drops [15], Yellepeddi and Palakurthi reviewed the advances in situ gel and nanoparticle eye drops [16], Alvarez-Trabado et al. reviewed the design of lipid nanoparticles for topical drug delivery [17], Clolkar et al. reviewed novel strategies for anterior segment ocular drug delivery [18], Madni et al. investigated non-invasive strategies in the posterior segment of the eye studies [19]. In contrast to these studies, we have not only investigated prodrugs, cyclodextrins, in situ gels and nanoparticle eye drops, but also investigated novel eye drop delivery systems in the application in anterior and posterior segments of the eye. More importantly, we have also studied high ocular bioavailability drops based on these novel delivery systems, such as prodrug-based self-assembled nanoparticle eye drops; cyclodextrin-based prodrug eye drops, in situ gel eye drops; nanoparticle-based cationic nanoparticle eye drops, surface-modified nanoparticle eye drops, nanoparticle eye drops with adhesion properties and ligand-targeted nanoparticle eye drops. The study of novel delivery systems in the anterior and posterior segments of the eye may provide a reference for the non-invasive treatment of ocular diseases and for the development of highly bioavailable eye drops. This section has been added to the introductory section.
Question2: Clinical application and commercial aspects are mentioned in the abstract but I feel that these are rather missing in most subsections and discussion is rather based on in vitro, ex vivo data. Please add some data where clinical trials or application can be found.
Response: Thank you for your kindly comments. New eye drop products already on the market: Lumigan® (prodrug), Travatan® (prodrug), Xalatan® (prodrug), Lotemax® (prodrug), Tiopex® (pH-sensitive in situ gel), Pilopine HS® (pH-sensitive in situ gel), Timoptic® GFS (ion-responsive in situ gel), Timoptic-XE (ion-responsive in situ gel) and studies in clinical trials: efficacy and safety of dexamethasone nanoparticles eye drops in diabetic macular edema (NCT05343156), chloroprocaine 3% gel eye drop as topical anestheticsin phacoemulsification (NCT04685538), study of brimonidine tartrate nanoemulsion eye drops in patients with ocular graft-vs-host disease (oGVHD) (NCT03591874), study of brimonidine tartrate nanoemulsion eye drop solution in the treatment of dry eye disease (DED) (NCT03785340) and effect of lipid based eye drops on tear film lipid layer thickness (NCT03399292) have been added to the section 5 novel eye drop products and clinical trials.
Question3: Nanoparticles are explained very much in detail which reflects on the current interest in such systems. However, I recommend the authors to be more comparative than only descriptive. E.g., section 4.4.3 should start with the advantages OVER other systems rather definition and general descriptions.
Response: Thanks lots. Each type of nanoparticle has its own unique advantages over other carriers. Liposomes are highly histocompatible and biodegradable and they can accommodate both hydrophilic and lipophilic components due to their unique structure. Solid lipid nanoparticles are solid particles that can avoid degradation of lipid components in liquid solutions, at the same times, they are easy to scale up for production. Polymeric nanoparticles are more stable and adhesive polymeric materials allow nanoparticles to adhere to mucous membranes. The amphiphilic nature of micelles makes them suitable for the delivery of both hydrophilic and lipophilic components. In addition, they are easy to manufacture and have a relatively small particle size compared to other nano carriers. Microemulsion can accommodate both hydrophilic and lipophilic components, and the formation of microemulsions generally does not require external forces, making the preparation process simple. Dendrimers have modifiable functional groups on their surfaces, which can be suitably modified to obtain suitable materials. The advantages of each nanoparticle in relation to other nanocarriers have now been added in the relevant sections.
Question4: Please explain more in detail: “Niosomes are more stable than liposomes due to the replacement of phospholipids with non-ionic surfactants”.I cannot see the point why phospholipids are less stable than non-ionic surfactants.
Response: Thanks to your comment. Natural phospholipids, which make up liposomes, contain unsaturated fatty acid chains in their molecules and are susceptible to oxidative hydrolysis to peroxides, dialdehydes, fatty acids and lysolecithin, resulting in reduced membrane fluidity and drug leakage. Non-ionic surfactants exist in solution in a non-ionic state and are less susceptible to the presence of strong electrolytes and less susceptible to acids and bases, making them highly stable. Niosomes are therefore more resistant to oxidative degradation and more stable than liposomes. This part of the explanation is also added in line 541.
Although the replacement of natural phospholipids by non-ionic surfactants makes Niosomes more chemically stable, an increase in irritation may follow. In addition, although liposomes containing natural phospholipids are susceptible to oxidation, these can be addressed for most liposomal prescriptions by the addition of antioxidants, for example.
Question5: In 4.4.7, I have doubts if (all) micoemulsions are thermodynamically stable. Please clarify.
Response: Thank you for your kindly comments. Microemulsion is a transparent or translucent, low viscosity, nanoscale droplet formed spontaneously from the aqueous phase, the oil phase and the surfactant in the appropriate proportions. Because no external work is required for the formation of microemulsions and the surfactant and co-surfactant act together to stabilise them, microemulsions are a thermodynamically stable system.
Question6: I really miss the explanation of current challenges both in subsections and also in the conclusion. It looks like this is a finished work but probably we are at only the beginning of the development of highly efficient formulations.
Response: Thanks lots. Novel eye drop delivery systems have shown improved efficacy in the treatment of anterior and posterior segment disorders of the eye, nevertheless, the negative phenomena associated with them should be taken into account. Prodrug may lead to increased lipid solubility and toxicity of the drug. Highly viscous in situ gels may lead to blurred vision. For some stimulus-sensitive materials with relatively weak gelation efficiency, the use of high concentrations of materials or combinations of several materials can increase their toxicity. Nanoparticles may suffer from low encapsulation rates, poor stability and a tendency for the drug to leak during storage. Therefore, not only the prescribing factors, stability and sterility of the product need to be considered in the early stages of eye drop development. Further research and development of methods to improve drug preparation and storage is also needed to ensure the efficacy and safety of eye drops. The current challenges faced by the new eye drops have been added to in the conclusions section.
Special thanks to you for your helpful comments!
We tried our best to improve the manuscript and made some changes in the manuscript. These changes which were marked in red font in the revised manuscript will not influence the content and frame work.
Thanks again for your kind help. We hope this revised version of our manuscript will be acceptable for publication. For any other questions, please feel free to contact us and we will response at our first time.
Sincerely yours,
Professor Chang-hong Wang
2022/5/17
Reviewer 3 Report
The manuscript describes the most innovative formulations of eye drop delivery systems for improving ophthalmic bioavailability of drugs. Extensive description of the different kinds of nanomaterials, their preparation and characterization in terms of efficacy in vitro and in vivo are reported, evidencing their advantages, properties and performance respect to conventional strategies. The manuscript is clear and fluent. The topics are covered in a comprehensive way and summary tables are a valuable tool for the reader and as a starting point for future researches. Only some typos are present and a few points need to be clarified. I suggest publication upon minor revision.
Specific comments:
Line 52. administrated, change into administered
Line 62. BAB, explain acronym the first time it appears in the text
Figure 1. please increase the size of the text and separate on one side the structure of the eye and routes of drug delivery and on the other side the ocular barriers, in order to make the figure more clear to the reader.
Figure 2. please increase the size of the text
Line 219. IOP, explain acronym the first time it appears in the text
Line 220. enables, change into enable
Line 236. thermal stability, when
Line 240. VEGFr2, change into VEGFR2
Line 264. but also were also, remove also
Table 2. prodrug eye drop in the title, change into cyclodextrin eye drop
Line 292. transformation, change into transform
Line 304. the more, change into the most
Figure 4. stimulates, change into stimulation
Line 313. the high critical solution temperature (UCST), change into upper
Line 404-405. list the nanoparticles in the same order as they are described in the text below
Figure 5. put the nanoparticles in the same order they are described in the text below, add O/W and W/O meaning in the figure legend
Line 418. conducted, change into concluded
Line 430. ibuprofen cationic, use of this molecule is not clear
Line 451. long-lasting, add way
Line 472. the comparison with doxorubicin solution is not clear
Line 488. are any examples of lipid drug couplers applied to eye drop reported in literature?
Line 500. loaeded, change into loaded
Line 502-506. this sentence is not clear
Line 610. significantly change into significant
Line 629. reduced dose, change into required reduced dose
Line 665. CsA, explain acronym the first time it appears in the text
Line 703. Polyamides, change into Poly(amidoamine)
Line 723. anionic DEX, explain acronym the first time it appears in the text
Line 745. table 3, change with table 4
Author Response
Dear Editors and Reviewers:
Thank you for your letter and for the reviewers' comments on our manuscript. We appreciate you taking the time to read and review it. The comments were valuable and helpful in revising and improving our paper. We have carefully studied these comments and have made corrections in the manuscript using a revision model and the corrections are marked in red, which we hope you will approve. The main corrections and point-by-point responses to the comments in the thesis are set out below.
Responds to the reviewer’s comments:
The manuscript describes the most innovative formulations of eye drop delivery systems for improving ophthalmic bioavailability of drugs. Extensive description of the different kinds of nanomaterials, their preparation and characterization in terms of efficacy in vitro and in vivo are reported, evidencing their advantages, properties and performance respect to conventional strategies. The manuscript is clear and fluent. The topics are covered in a comprehensive way and summary tables are a valuable tool for the reader and as a starting point for future researches. Only some typos are present and a few points need to be clarified. I suggest publication upon minor revision.
Specific comments:
Question1: Line 52. administrated, change into administered
Response: Thanks to your comment. The spelling mistake of ‘administrated’ has been corrected as administered (Line52).
Question2: Line 62. BAB, explain acronym the first time it appears in the text
Response: Thank you for your kindly comments, the full definition of the BAB, blood-aqueous barrier, has been added (Line 62).
Question3: Figure 1. please increase the size of the text and separate on one side the structure of the eye and routes of drug delivery and on the other side the ocular barriers, in order to make the figure more clear to the reader.
Response: Thanks lots. The structure of the eye has been placed on the left side of Figure 1 and the ocular barrier on the right side of Figure 1. At the same time, the size of the text has been increased.
Question4: Figure 2. please increase the size of the text
Response: Thank you very much! The size of the text in Figure 2 has been increased.
Question5: Line 219. IOP, explain acronym the first time it appears in the text
Response: Thanks lots. The full name of the IOP, intraocular pressure, has been added (Line 219).
Question6: Line 220. enables, change into enable
Response: Thanks to your comment, enables has been revised into enable (Line 224).
Question7: Line 236. thermal stability, when
Response: Thanks lots, revision have been completed (Line 229).
Question8: Line 240. VEGFr2, change into VEGFR2
Response: Thank you very much! VEGFr2 (Line 232) was written correctly, which can be found with reference (https:// doi: 10.1021/jm7011276).
Question9: Line 264. but also were also, remove also
Response: Thank you for your kindly comments, also has been removed (Line 292).
Question10: Table 2. prodrug eye drop in the title, change into cyclodextrin eye drop
Response: Thanks lots, prodrug has been changed into cyclodextrin in Table 2.
Question11: Line 292. transformation, change into transform
Response: Thanks lots, transformation has been revised into transform (Line 315).
Question12: Line 304. the more, change into the most
Response: Thank you very much! The more has been change into the most (Line 327).
Question13: Figure 4. stimulates, change into stimulation
Response: Thanks to your comment, stimulates has been revised into stimulation in Figure 4.
Question14: Line 313. the high critical solution temperature (UCST), change into upper
Response: Thanks lot, the high critical solution temperature (UCST) has been changed into the upper critical solution temperature (UCST) (Line 336).
Question15: Line 404-405. list the nanoparticles in the same order as they are described in the text below
Response: Thank you for your kindly comments. The nanoparticles have been listed in the same order as they are described in the text in Figure 5 (Line 453-454).
Question16: Figure 5. put the nanoparticles in the same order they are described in the text below, add O/W and W/O meaning in the figure legend
Response: Thanks to your comment. The nanoparticles has been listed in the same order as they are described in the text in Figure 5 and the full definition of O/W (oil-in-water) and W/O (water-in-oil) has been added in Figure 5.
Question17: Line 418. conducted, change into concluded
Response: Thanks lots, conducted has been revised into concluded (Line 471).
Question18: Line 430. ibuprofen cationic, use of this molecule is not clear
Response: Thanks to your comment, ibuprofen cationic was incorrectly stated, it should be a cationic liposome loaded with ibuprofen (Line 487).
Question19: Line 451. long-lasting, add way
Response: Thank you very much! The way has been added (Line 519).
Question20: Line 472. the comparison with doxorubicin solution is not clear
Response: Thanks lots. The duration of reduction of intraocular pressure after topical drops of dorzolamide solution and carbomer-coated niosomes was approximately 1.5 h and 6 h. Thus, carbomer-coated niosomes were four times more effective in reducing pressure than dorzolamide solution. The duration of lowering of dorzolamide solution has been added to the article (Line 541).
Question21: Line 488. are any examples of lipid drug couplers applied to eye drop reported in literature?
Response: Thank you for your kindly comments. Lipid drug conjugates (LDC) are developed by converting a hydrophilic drug to a lipophilic drug conjugate or a prodrug by the addition of an ester or amide group. (https:// doi:10.1016/b978-0-323-46143-6.00016-6). As a novel drug delivery system, it has been reported in the form of eye drops, for example, Stella et al. obtained the LDC drug 4´-trisnorsqualenoylacyclovir (SQACV) by covalently linking the 4´-hydroxyl group of acyclovir to the isoprene chain of squalene and subsequently formulated SQACV as a nonpolymeric nanoassemblies by nanoprecipitation. Pharmacokinetic characterisation in rabbit tear fluid and aqueous humour showed that the SQACV nanoassemblies increased the ACV content in rabbit aqueous humour compared to the free ACV eye drops. This suggests that LDC is a very promising tool to increase the ocular bioavailability of drugs. As this section was not mentioned in the original manuscript, it has now been added in the revised version (Line 572-577).
Question22: Line 500. loaeded, change into loaded
Response: Thanks to your comment, loaeded has been changed into loaded (Line 570)
Question23: Line 502-506. this sentence is not clear
Response: Thank you very much! The sentence was unclear in its description of the content and has been changed to Liu et al. developed coumarin-6 loaded NLC (C6-NLC) using melt emulsion technolo-gy, followed by surface modification with chitosan-hydrochloride (CH) and chi-tosan-N-acetylcysteine (CS-NAC), respectively. The result of in vitro mucosal adhesion demonstrated that the presence of CS-NAC on the C6-NLC surface provided the most obvious enhancement in adhesion due to the formation of both non-covalent (ionic) and covalent (disulfide bridges) interactions with mucus chains, and the higher the concentration of CS-NAC, the longer the retention time of the nanoparticles in the oc-ular tissue. In addition, corneal penetration results showed that CS-NAC-NLC parti-cles were able to penetrate the entire corneal epithelium mainly via a transcellular pathway (Line585-593).
Question24: Line 610. significantly change into significant
Response: Thank you for your kindly comments. the word of significantly has been changed into significant (Line 714).
Question25: Line 629. reduced dose, change into required reduced dose
Response: Thanks lots. The reduced dose has been changed into required reduced dose (Line742).
Question26: Line 665. CsA, explain acronym the first time it appears in the text
Response: Thanks to your comment. The full name of the CsA, cyclosporine A, was first appeared on line 686.
Question27: Line 703. Polyamides, change into Poly(amidoamine)
Response: Thank you very much! Polyamides has been changed into Poly(amidoamine) (Line 814).
Question28: Line 723. anionic DEX, explain acronym the first time it appears in the text
Response: Thanks lots. The full name of the DEX, dexamethasone, was first appeared on line 263.
Question29: Line 745. table 3, change with table 4
Response: Thanks to your comment, table 3 has been changed into table 4.
Special thanks to you for your helpful comments!
We tried our best to improve the manuscript and made some changes in the manuscript. These changes which were marked in red font in the revised manuscript will not influence the content and frame work.
Thanks again for your kind help. We hope this revised version of our manuscript will be acceptable for publication. For any other questions, please feel free to contact us and we will response at our first time.
Sincerely yours,
Professor Chang-hong Wang
2022/5/15
Reviewer 4 Report
Dear Editor, in the submitted review the research progress of novel eye drop delivery systems, was extensively presented. The review is well organized and almost all used works in this area have been adequately presented. For this reason I will propose to accept it for publication. Some minor comments. In polymeric nanoparticles chitosan is used also extensively for eye drops formulations, due to its mucoadhesive and antibacterial properties and some review have been also published in this area. However, these formulations are not reported and I propose to include them in revised manuscript (please see Pharmaceutics 2020, 12, 594; http://doi:10.3390/pharmaceutics12060594 and Polymers 12, 1519; 2020, https://doi:10.3390/polym12071519.)
Author Response
Dear Editors and Reviewers:
Thank you for your letter and for the reviewers' comments on our manuscript. We appreciate you taking the time to read and review it. The comments were valuable and helpful in revising and improving our paper. We have carefully studied these comments and have made corrections in the manuscript using a revision model and the corrections are marked in red, which we hope you will approve. The main corrections and point-by-point responses to the comments in the thesis are set out below.
Responds to the reviewer’s comments:
Question: Dear Editor, in the submitted review the research progress of novel eye drop delivery systems, was extensively presented. The review is well organized and almost all used works in this area have been adequately presented. For this reason I will propose to accept it for publication. Some minor comments. In polymeric nanoparticles chitosan is used also extensively for eye drops formulations, due to its mucoadhesive and antibacterial properties and some review have been also published in this area. However, these formulations are not reported and I propose to include them in revised manuscript (please see Pharmaceutics 2020, 12, 594; https://doi:10.3390/pharmaceutics12060594 and Polymers 12, 1519; 2020, https://doi:10.3390/polym12071519.)
Response: Thank you for your kindly comments. The adhesion, antibacterial activity and pro-permeation properties of chitosan (CS) make it one of the most frequently chosen polymeric nanoparticles for ocular drug delivery systems. Chitosan nanoparticles were not described in detail in the original manuscript and have now been added in the revised manuscript.
The details are as follows: CS and its derivatives are commonly used as materials for adherent polymeric nanoparticles because of their adhesion properties, antibacterial activity, and ability to facilitate drug penetration by opening tight intercellular junctions [161]. CS-nanoparticles loaded with carteolol (CRT) were prepared by Ameeduzzafar et al. with a particle size of 243 nm, drug loading of 49.21 ± 2.73%, and entrapment efficiency of 69.57 ± 3.54%. In vitro release studies showed a sustained release for 24 h as compared to drug solution, and scintigraphy study demonstrated good spread and retention in the pre-corneal area as compared to the aqueous CRT solution and prolonged reduction in intraocular pressure [162]. Karava et al. found that pure CS and its derivatives with 2-acrylamido-2-methyl-1-propanesulfonic acid (AAMPS) and [2-(methacryloyloxy)ethyl]dimethyl-(3-sulfopropyl)ammonium hydroxide (MEDSP) could both be used as carriers for polymeric nanoparticles for intraocular delivery of dexamethasone sodium phosphate (DxP) and chloramphenicol (CHL) [163]. In addition to acting as a backbone material for polymeric nanoparticles, CS and its derivatives can also be coated on the surface of nanoparticles to impart adhesion properties to the nanoparticles. For example, polylactic acid nanoparticles loaded with rapamycin had good retention of nanoparticles on the corneal surface when coated with chitosan [164].
Special thanks to you for your helpful comments!
We tried our best to improve the manuscript and made some changes in the manuscript. These changes which were marked in red font in the revised manuscript will not influence the content and frame work.
Thanks again for your kind help. We hope this revised version of our manuscript will be acceptable for publication. For any other questions, please feel free to contact us and we will response at our first time.
Sincerely yours,
Professor Chang-hong Wang
2022/5/15
Round 2
Reviewer 1 Report
The review has been completely revised and new and more significant references have been added.